# Summertime Rossby waves in climate models: Substantial biases in surface imprint associated with small biases in upper-level circulation

Fei Luo[1,2], Frank Selten[2], Kathrin Wehrli[3], Kai Kornhuber[4,5], Philippe Le Sager[2], Wilhelm May[6], Thomas Reerink[2], Sonia I. Seneviratne[3], Hideo Shiogama[7], Daisuke Tokuda[8], Hyungjun Kim[8,9,10], and Dim Coumou[1,2]

[1]Institute for Environmental Studies (IVM), VU University Amsterdam, Amsterdam, Netherlands
[2]Royal Netherlands Meteorological Institute (KNMI), De Bilt, Netherlands
[3]Institute for Atmospheric and Climate Science, Department of Environmental Systems Science, ETH Zurich, Zurich, Switzerland
[4]Earth Institute, Columbia University, New York, United States
[5]Lamont-Doherty Earth Observatory, Columbia University, New York, United States
[6]Centre for Environmental and Climate Science (CEC), Lund University, Lund, Sweden
[7]Center for Global Environmental Research, National Institute for Environmental Studies, Tsukuba, Japan
[8]Moon Soul Graduate School of Future Strategy, Korea Advanced Institute of Science and Technology, Daejeon, Korea
[9]Department of Civil and Environmental Engineering, Korea Advanced Institute of Science and Technology, Daejeon, Korea
[10]Institute of Industrial Science, University of Tokyo, Tokyo, Japan

*Correspondence to*: Fei Luo (fei.luo@vu.nl)

**Abstract.** In boreal summer, circumglobal Rossby waves can promote stagnating weather systems that favor extreme events like heatwaves or droughts. Recent work showed that amplified Rossby wavenumber 5 and 7 show phase-locking behavior which can trigger simultaneous warm anomalies in different breadbasket regions in the Northern Hemisphere. These types of wave patterns thus pose a potential threat to human health and ecosystems. The representation of such persistent wave events in summer and their surface anomalies in general circulation models (GCMs) has not been systematically analyzed. Here we validate the representation of wavenumbers 1-10 in three state-of-the-art global climate models (EC-Earth, CESM, and MIROC), quantify their biases and provide insights into the underlying physical reasons for the biases. To do so, the ExtremeX experiments output data were used, consisting of (1) historic simulations with a freely running atmosphere with prescribed ocean, and experiments that additionally (2) nudge towards the observed upper-level horizontal winds, (3) prescribe soil moisture conditions, or (4) do both. The experiments are used to trace the sources of the model biases to either the large-scale atmospheric circulation or surface feedback processes. Focusing on wave-5 and wave-7, we show that while the wave's position and magnitude are generally well represented during high amplitude (> 1.5 s.d.) episodes, the associated surface anomalies are substantially underestimated. Near-surface temperature, precipitation and mean sea level pressure are typically underestimated by a factor of 1.5 in terms of normalized standard deviations. The correlations and normalized standard deviations for surface anomalies do not improve if the soil moisture is prescribed. However, the surface biases are almost

entirely removed when the upper-level atmospheric circulation is nudged. When both prescribing soil moisture and nudging the upper-level atmosphere, then the surface biases remain quite similar to the experiment with nudged atmosphere only. We conclude that the near-surface biases in temperature and precipitation are in the first place related to biases in the upper-level circulation. Thus, relatively small biases in the models' representation of the upper-level waves can strongly affect associated temperature and precipitation anomalies.

## 1 Introduction

The past decade has witnessed a series of unprecedented boreal summer weather extreme events around the globe, such as the 2010 Russian heatwave, 2012 North American heatwave, and the record-breaking heatwaves of 2015, 2018 and 2019 in Europe (Barriopedro et al., 2011; Kornhuber et al., 2019; Krzyżewska and Dyer, 2018; Wang et al., 2014; Huntingford et al., 2019; Xu et al., 2021). Some of these events also happened simultaneously with other types of extremes such as the persistent Russian heatwave and Pakistan flood in 2010 July and August (Lau and Kim, 2012; Martius et al., 2013). Persistent weather extremes are often induced by certain Rossby wave patterns. For instance, Recurring Rossby Waves Packets (RRWPs) can lead to cold spells in winter and hot spells in summer (Röthlisberger et al., 2019). These persistent weather extremes can have disastrous impacts on human health and societies, such as wide spread crop failure, infrastructure damage and properties loss, especially when they co-occur (Zscheischler et al., 2018). Several other studies have identified that amplified circumglobal waves favor the occurrence of weather extremes in specific regions (Screen and Simmonds, 2014; Kornhuber et al., 2020). Specifically, in summer wave-5 (Ding and Wang, 2005; Kornhuber et al., 2020) and wave-7 (Kornhuber et al. 2019., 2020) have preferred phase positions and thereby favor simultaneous extremes in major breadbasket regions (Kornhuber et al., 2020).

Several mechanisms can promote quasi-stationary Rossby waves including strong convective forcing from monsoons (Di Capua et al., 2020), extratropical sea surface temperature (SST) anomalies (McKinnon, et al., 2016; Vijverberg, et al., 2020), soil moisture anomalies (Teng and Branstator, 2019), waveguide effect (Hoskins and Ambrizzi, 1993), and wave-resonances (Petoukhov et al., 2013, 2016; Kornhuber et al., 2017; Thomson and Vallis, 2018). Recent work by Di Capua et al. (2020) found that the latent heat release during the Indian Summer Monsoon initiates a circumglobal teleconnection pattern, which reflects a wave-5 type pattern in the northern mid-latitudes. Extratropical SSTs can interact with atmospheric waves creating quasi-stationary atmospheric Rossby waves favorable for e.g. hot days in the eastern United States (McKinnon et al., 2016; Vijverberg and Coumou, 2022). Moreover, waves can be excited by reduced soil moisture and then maintained by waveguides in the Northern Hemisphere mid-latitudes (Teng et al., 2019). Quasi-Resonant Amplification (QRA) theory suggests that synoptic scale Rossby waves can be trapped within the mid-latitude waveguides, where they can get amplified given suitable forcing conditions (Petoukhov et al. 2013). Since the wave's energy is not lost via meridional dispersion, waves tend to propagate over long longitudinal distances and can sometimes form a circumglobal wave pattern (Hoskins and Ambrizzi, 1993; Branstator, 2002b; Teng and Branstator, 2019).

Climate models are important tools for process understanding and assessment of future climate risks. However, most of the previous studies that link specific Rossby wave patterns to regional extreme events are based on reanalysis or observational data. Although studies such as Garfinkel et al. (2020); Wills et al. (2019) have analyzed waves in models, their focus is not on summer. Moreover, no study has analyzed the phase-locking behavior of amplified, quasi-stationary Rossby waves in summer, despite the risks these waves can create for society. Furthermore, most studies have not analyzed waves above wave number 6. Studies by Branstator et al. (2002 & 2017) have also looked into models but focused on seasonal means and/or winter. A multi-model validation study of quasi-stationary Rossby waves in boreal summer is still lacking. Another key issue here is the general underestimation of models in atmospheric blocking in summer (Davini and D'Andrea 2020), likely linked to a misrepresentation of the processes that maintain blocking. This reduces the reliability of future model projections, in particular for extreme weather events (Scaife et al. 2010; Shepherd 2014). A recent study by Davini and D'Andrea (2020) analyzed the representation of both winter and summer blocking in models from the Coupled Model Intercomparison Project Phase 3 (CMIP3, 2007), CMIP5 (2012), and CMIP6 (2019). Although biases in CMIP6 models were reduced by 50% compared to CMIP3, in some key regions like Europe, the biases still remain. Thus, even CMIP6 models cannot truthfully reproduce blocking frequencies over Europe nor capture the observed significant increase in summertime blocking over Greenland (Davini and D'Andrea 2020).

Furthermore, while here we focus on wave-5 and 7, extreme events can also occur with other wavenumbers. High-amplitude slow-moving planetary waves are associated with persistent surface weather conditions. For example, QRA mechanism can explain the generation of circumglobal Rossby waves with wavenumbers 6 to 8 in the Northern Hemisphere (Coumou et al., 2014). The evidence for QRA in the Southern Hemisphere (SH) is also found to exist for wavenumbers 4 and 5 (Kornhuber., 2017). Whether climate models can also reproduce this wide range of wave numbers and their associated surface anomalies needs to be tested.

Thus, to increase confidence in future projections of extreme summer weather, a validation of state-of-art climate models in their representation of quasi-stationary Rossby waves in summer is essential. We analyze the upper-level dynamical characteristics, in terms of the wave's amplitude and phase position, as well as the associated anomalies in surface variables. We systematically validate the representation of summertime Rossby waves in three state-of-the-art climate models, focusing on wave-5 and 7 in the northern mid-latitudes, their phase-locking behavior and surface anomalies. Further, by nudging the upper-level atmosphere or prescribing soil-moisture, we aim to understand the origin of the biases in surface variables during high-amplitude wave episodes.

This paper aims at addressing the following questions:

1: Can models capture the key characteristics of high-amplitude quasi-stationary Rossby waves in summer?

2: What are the near-surface temperature, precipitation, and mean sea level pressure anomalies from such waves and how do they compare to observations?

3: Do potential model biases in surface variables originate primarily from the atmospheric circulation or from land surface-feedbacks?

## 2 Data and Methods

### 2.1 ExtremeX experiment

We use simulation output from three Earth System Models (ESM) that participated in the ExtremeX modeling experiment (Wehrli et al., 2021, in review): European Community Earth System Model version 3.3.1 (EC-Earth 3.3.1; Döscher et al., 2021, in review), Community Earth System Model version 1.2 (CESM1.2; Hurrell et al., 2013), and Model for Interdisciplinary Research on Climate version 5 (MIROC5; Watanabe et al., 2010). The configuration of CESM and MIROC was used for CMIP5 (CMIP5; Taylor et al., 2012), whereas EC-Earth is the latest 3$^{rd}$ generation model which was used for CMIP6 (Eyring et al., 2016). The ExtremeX modeling experiments were designed to disentangle the influence from atmospheric dynamics vs. soil moisture feedback on extreme events such as heatwaves, droughts, and other extremes. By nudging either the upper-level atmosphere or prescribing the soil moisture state, or both, the individual effects can be compared across different models. Details on the experimental set-up and atmospheric nudging approach are described in a recent study which examined five individual heatwaves in the period of 2010 – 2016 (Wehrli et al., 2019).

### 2.2 Model data output

Here we use four out of five sets of simulations from ExtremeX, which are all run in Atmospheric Model Intercomparison Project (AMIP) (Gates et al., 1999) style with prescribed monthly mean SSTs and sea-ice. The four experiments are run with respectively (1) interactive atmosphere and soil moisture, as control simulation (AISI), (2) nudged atmosphere (mostly above 700 hPa) but interactive soil moisture (AFSI), (3) nudged atmosphere with prescribed soil moisture (AFSF), and (4) interactive atmosphere with prescribed soil moisture (AISF). The experiment period extends from January 1979 to December 2016 for both EC-Earth and CESM and till December 2015 for MIROC. Overall output is provided 6-hourly on different model grids (in number of grid points longitude x latitude): EC-Earth (512 x 256), CESM (288 x 192), and MIROC (256 x 128). There are five ensemble members for the free atmosphere experiments (AISI and AISF) for the full period. However, for the nudged atmosphere experiments (AFSI and AFSF) only one simulation was sufficient. All model and reference data are regridded to the same resolution (256 x 128) for comparisons.

## 2.3 Atmospheric nudging

To constrain the natural variability in large-scale atmospheric circulation, a grid-point nudging method was implemented in the AFSI and AFSF experiments (Jeuken et al., 2016). This approach forces the atmospheric large-scale circulation by introducing a tendency term in the horizontal wind components. By taking the differences between the model simulations and reference dataset, the added tendency term is computed. Kooperman et al. (2012) demonstrated that when the horizontal wind is nudged towards a reference state, the impact of natural variability is substantially reduced. The strength of the nudging can be modified by a relaxation time scale, which was chosen to be 6h following other studies (Kooperman et al., 2012). All models use 6-hourly wind field data from the ERA-Interim reanalysis as reference data (Dee et al., 2011). The vertical profile of the atmospheric nudging (See Appendix Figure B1) shows that nudging starts around 700hPa but only with a very weak nudging strength. The nudging strength increases gradually in the upward direction and full nudging is only applied above ca. 400hPa. Thus, the mid-to-upper atmosphere is nudged, and it is important to note that the planetary boundary layer is free to adjust in the nudged experiments.

## 2.4 ERA5 reanalysis data

For the study period 1979 to 2016, weekly meridional wind data at 250 hPa (v250), near-surface temperature (t2m), and mean sea level pressure (mslp) are taken from the ERA5 reanalysis for the summer months June, July, and August (JJA) (Hersbach et al., 2020). For precipitation (prcp), land-only data is used from bias-adjusted ERA5 (WFDE5_CRU, Cucchi et al., 2020). Also, weekly t2m data is detrended to its climatological mean (1979 – 2016) values of that week.

## 2.5 Extracting circumglobal waves and phase-locking analysis

High-amplitude wave episodes are selected based on the Fourier transformation analysis of weekly-mean v250 averaged over 35N to 60N, both in ERA5 and models analogous to previous studies (Kornhuber et al. 2019, 2020). Wave events are identified as those weeks with wave amplitudes higher than 1.5 standard deviations (s.d.) of the climatology calculated from 494 weeks (38 years times 13 summer weeks per year) for ERA5. Since the AISI and AISF runs both have 5 ensemble members, the total numbers of weeks are as follows: 494 x 5 = 2470 for EC-Earth and CESM, and 481 x 5 = 2405 for MIROC. For AFSI and AFSF, only one member is used for each model. Then, the composite surface imprints of near-surface temperature, precipitation, and mean sea level anomalies were obtained from those wave episode periods. By imprint we understand anomalies in variables at various atmospheric levels, as well as anomalies in the surface variables, occurring during high-amplitude (over 1.5 s.d.) events of waves 5 and 7. We note here, that the subset of high-amplitude events in free-running experiments may differ significantly from events in ERA5 and nudged experiments, hence, some imprints may be associated with processes not directly related to wave amplification. To analyze preferred phase positions of individual waves, we determine the probability density functions of phase position for high-amplitude wave episodes.

**2.6 Model bias definition**

Our model experiments can be used to better understand the source of the models' biases in surface anomalies, and analyze whether those are primarily coming from the upper-level atmosphere or from the land surface component. To do so, we compare the biases in our different modelling experiments. Here we define 'bias' as the difference between a model composite and a reanalysis composite. Using our different model setups, we can then define specific biases for various variables: the total bias (B_tot), the bias from the upper-level atmospheric circulation (B_atm), the bias from land-atmosphere interactions (B_land), and the remaining residual bias (B_res).

In the AISI experiment, both the atmosphere and land surface component are allowed to interact and evolve freely, and this experiment thus defines the total bias of surface variables:

B_tot  =  AISI – ERA5

When prescribing soil moisture in AISF, we assume that the bias originating from the land-component is effectively removed and only the bias from the atmosphere acting upon near-surface variables remains. Thus:

B_atm =  AISF – ERA5

In contrast, when nudging the upper-level atmosphere, the upper-level circulation pattern is constrained in the model and thus the bias arises from land-atmosphere interactions.

B_land =  AFSI – ERA5

When both nudging the upper-level atmosphere and prescribing soil moisture, the biases in surface variables are expected to be strongly reduced with only a residual bias remaining:

B_res =  AFSF – ERA5

**3 Results**

**3.1 Climatology of summertime Rossby waves**

We first assess whether the climate models are able to represent the mean state in terms of wave amplitude and variability for wavenumbers 1-10 in June, July, and August. Figure 1 compares wave spectra for wavenumbers 1 to 10 from the AISI

experiment with those of the ERA5 reanalysis. Overall, the wave amplitudes, regardless of wave numbers and models, are
reasonably well reproduced with errors in the model climatology ranging from 5% (wave-10) to 12% (wave-3). This also
applies to the variance in wave amplitude as given by the whisker bars for each model at different wavenumbers. For all
models, the wave amplitudes and variabilities follow the same behavior with increasing values from wave number 1 to 5 and
decreasing values from wavenumber 6 to 10. ERA5 shows the peak for both the wave amplitude and variance at wavenumber
6, which might suggest a systematic bias in the models, or alternatively it might be an under-sampling issue in ERA5.

**3.2 Wave phase-locking behaviors**

Following Kornhuber et al (2020), we use 1.5 s.d. above the mean wave amplitude as a threshold to define high-amplitude
wave events to analyze phase-locking behavior of high-amplitude waves 5-8. The phase positions of high amplitude episodes
are shown in Fig. 2. In ERA5 waves have inherent phase-locking properties, especially for waves 5 and 7, as visible by a single
peak in the probability density function. This is consistent with the work from Kornhuber et al. (2019) who used different
reanalysis data, i.e. NCEP-NCAR (Kalnay et al., 1996). Also, in the NCEP-NCAR reanalysis data, waves 6 and 8 do not really
show a preferred phase position. In our experiments, across all three models, strong phase-locking behavior is detected for
wave-7. For wave-5, two models (CESM and MIROC) show phase locking that is comparable to ERA5, however EC-Earth
underestimates the peak, and thus the strength of phase-locking.

ERA5 shows no phase-locking for wave 6 and wave 8 but rather only a mild preference for some phase positions. The models
capture this, with only MIROC showing fairly pronounced phase-locking behavior for wave 6.  Detailed histograms comparing
the models with ERA5 are provided in Fig. B2.

**3.3 High-amplitude wave episodes and their surface imprint**

We find that the preferred phase position of waves 5 and 7 is reasonably well represented in models, we next analyze high-
amplitude episodes (i.e. those exceeding 1.5 s.d.) in more detail. The wave episode occurrence is computed as the percentage
of weeks showing high amplitude wave episodes during the 1979 to 2015/16 summer. For ERA5, this is the case for 8.1% and
7.1% of all weeks for wave-5 and wave-7 events, respectively. The occurrence-frequencies are quite comparable for the
models: 7.7% (EC-Earth), 8.0% (CESM), and 7.9% (MIROC) for wave-5, and 8.1% (EC-Earth), 7.8% (CESM), and 8.0%
(MIROC) for wave-7. Figures 3 and 4 show the upper-level meridional wind (v at 250 hPa, absolute field), and anomalies of
near-surface temperature (t2m), precipitation (prcp), and sea level pressure (mslp) during high-amplitude events in ERA5
(panel a). The same variables are shown for the free-running atmosphere and soil moisture (AISI) experiments using the three
climate models (panels b-d). Extended analysis for other wavenumbers revealed that the evaluated models are capable of
reproducing the high-amplitude Rossby waves 4 to 8 and their associated surface anomalies reasonably well (Fig. B3 - B5).
The results imply that the model is able to reproduce summertime surface anomalies associated to different wavenumber
episodes. Using anomalies for the upper-level meridional winds, in contrast to absolute v250, gives  consistent results (compare

Fig. 3 and 4 and Fig. B10 and B11, respectively). Furthermore, Fig. B12 and B13 show that the composite anomalies for v250, t2m, and mslp are significant for both waves 5 and 7, accounting also for the False Discovery Rate (FDR; Benjamini & Hochberg, 1995).

Additionally, to quantify the bias of all models and to visualize how close the models are to the ERA5 reanalysis data, we present our findings in a Taylor Diagram (Taylor 2001) for wave-5 (Fig. 5) and wave-7 (Fig. 6). A Taylor Diagram presents three key statistics in one single plot: the Pearson correlation between the observed and modeled spatial pattern; the centered Root Mean Square Error (RMSE) of the modeled field compared to observed; and the normalized spatial standard deviation of the modeled field, as compared to observed. Thus, v250 during high-amplitude wave episodes, as well as t2m, prcp, and

mslp anomalies from the different models are plotted in the Taylor Diagram, showing their correspondence to ERA5 reanalysis data.

All models are able to capture the mean upper-level circulation patterns reasonably well for wave-5 with correlations for v250 of 0.86 (EC-Earth), 0.95 (CESM), and 0.88 (MIROC) (Fig. 5 & Table A1). This is consistent with our findings from Fig. 1

and 2. For the wind speed anomalies, CESM and MIROC have similar magnitudes compared to ERA5 data, whereas the signal from EC-Earth is weaker. The n.s.d. for EC-Earth is 0.70, for CESM it is 1.04. and for MIROC it is 1.24. This also holds for t2m anomalies during wave-5 events, as all models are able to reproduce the patterns found in ERA5, such as the continental-scale patterns of positive and negative temperature anomalies for central North America (+), western Europe (-), and central Europe (+). However, the strength of the patterns is weaker in EC-Earth, especially for eastern Eurasia. The correlations of the

t2m anomalies are substantially smaller for EC-Earth (0.55) and MIROC (0.48) than for CESM (0.81). As for prcp, all model correlations are below 0.50 with MIROC showing the lowest correlation (0.18), followed by CESM (0.43) and EC-Earth (0.46). The correlation values for mslp in models vary from 0.52 (MIROC), 0.58 (EC-Earth) to 0.80 (CESM). As for the multi-model mean (MMM) n.s.d. of the different examined variables, there is a decline from v250 (0.99) to the surface variables t2m (0.71), mslp (0.69) and prcp (0.63). Thus, surface anomalies are consistently too weak. Both reanalysis data and models show

strong positive anomalies in sea level pressure in the eastern basin of the Atlantic Ocean (west coast of Europe) during wave-5 episodes (Fig. 3).

For wave-7 episodes, the upper-level circulation patterns compare well to ERA5 data as shown by the field correlations in Table A1: 0.84 (EC-Earth), 0.84 (CESM), and 0.82 (MIROC). Again, this confirms that the models give a satisfactory

performance in producing correct upper-level circulation patterns during high-amplitude wave episodes. This thus again indicates that models capture phase-locking well. Correlations for surface variables are lower, with t2m correlation of 0.70 (EC-Earth), 0.63 (CESM), and 0.53 (MIROC). The positive t2m anomalies are quite pronounced in the regions of central North America, western Europe, northern Europe and central Eurasia. All models are able to reproduce the sign of t2m anomalies in these regions but with smaller magnitudes than in ERA5. The n.s.d. values of the t2m anomalies are 0.62 (EC-

Earth), 0.61 (CESM), and 0.67 (MIROC) (Table A2). The large-scale precipitation anomaly patterns in EC-Earth relate well to WFDE5_CRU data for North America, whereas in both CESM and MIROC there is more noise. The correlation of precipitation anomalies is 0.32 in the MMM. The large-scale patterns of mslp during wave-7 episodes match relatively well with ERA5 data with a MMM correlation value of 0.63. At the eastern side of the Atlantic Ocean (west coast of Europe), strong negative mslp anomalies can be seen for the wave-7 composites, which is opposite to the wave-5 signal here. Also, positive mslp anomalies are found during wave-7 episodes at the east coast of North America (Fig. 4), whereas the location shows negative anomalies during wave-5 episodes (Fig. 3).

One common finding from both wave-5 and wave-7 episodes is that the biases (n.s.d. $\geq 0.75$) in upper-level circulation are smaller compared to more-pronounced biases in the surface anomalies t2m, prcp, and mslp. All models substantially underestimate the magnitude of t2m, prcp, and mslp anomalies associated with wave-5 and wave-7 episodes, typically by a factor of 1.5 (Table A4).

### 3.4 Investigating sources of model biases

Next, we aim to infer the biases from composites of anomalies shown in Figs 3 and 4 for the upper-level wind and surface fields. As defined in the methods section, the bias maps were computed as the differences between the selected variables' anomalies in the models and in the reanalysis data ERA5 during high-amplitude wave-5 and wave-7 events. Note that the biases that we refer to in surface variables are the biases of the anomalies instead of the absolute bias of the models. Here we present and describe the EC-Earth bias maps only. Equivalent plots for the other two models, with qualitatively similar outcomes, can be found in Fig. B6 to Fig. B9.

Here, we also employ the different nudged experiments: AISF (soil moisture prescription), AFSI (upper-level atmosphere nudging), and AFSF (nudging both) (see data section above for details). Overall, when nudging both atmosphere and soil moisture, the residual bias B_res is, as expected, negligible. This is true for both wave-5 and wave-7 episodes in all models and all analyzed variables (Fig. 7 & Fig. 8).

By nudging the atmosphere, the bias from the atmospheric part (B_atm) is (of course) almost completely removed for the v250 anomaly across all models (see Fig. 7(a), B_land). More interestingly, Fig. 7(b) shows that most of the EC-Earth t2m anomaly bias is also removed when we nudge the upper-level atmosphere. Thus, the total bias (B_tot) in t2m is almost completely explained by the upper-level atmospheric bias (B_atm), and the contribution of the land-atm bias (B_land) is negligible (Fig 7(b)).

As for wave 5, the free-running EC-Earth (AISI) has a relatively smaller bias in v250 (blue square in Fig. 5(a)) with a correlation of ~0.9, a RMSE of ~0.5 and a n.s.d. of ~0.7 compared to the biases in the surface variables. In other words, the

pattern is very similar but with a somewhat underestimated strength in terms of wind speed. Still the bias in t2m (B_tot, blue square in Fig. 5 (b)) is substantially larger with correlation of ~0.6, RMSE ~0.9, and n.s.d ~0.6. Thus, the t2m anomaly is

underestimated with about a factor 1.7 (n.s.d ~0.6). This substantial bias in t2m is almost completely removed when nudging the upper-level wind field, i.e. removing the bias in v250. This is shown by the blue triangle in Fig. 5 (b) for AFSI which has a correlation of 0.94, a RMSE of 0.36 and a n.s.d. of 1. As can be seen in Fig. 5, the other models behave qualitatively in a similar way, with substantial biases in near-surface temperature (in AISI, squares in Fig. 5(b)) which are largely removed when the bias in upper level wind is removed by nudging (in AFSI, triangles in Fig. 5(b)).


Errors in precipitation anomalies are not fully removed when nudging upper-level circulation. Fig. 5 (c) shows some reduction in the overall magnitude of errors in precipitation, in particular the field correlation and n.s.d. improve by almost a factor 2 (Fig. 5(c)). Still the RMSE only marginally reduces when upper-level atmospheric nudging is applied. Biases in sea surface pressure anomalies, on the other hand, are almost completely removed with nudging (Fig. 5(d)).


Similarly, for wave-7 episodes, Fig. 8 (b) confirms our finding that nudging the upper-level atmosphere alone reduces the bias in surface temperature dramatically. Therefore, the total bias (B_tot) in t2m can be explained to a large extent by B_atm, and again the land contribution to the total bias is insignificant (Fig. 8(b)). Specifically, with the aid of a Taylor Diagram (Fig.6 (b) & Fig.6 (d)), there is a clear improvement when nudging the atmosphere (AFSI) as compared to the control run AISI and

the prescribed soil moisture run AISF. The t2m bias still remains substantial when prescribing the soil moisture. The actual n.s.d and RMSE values for v250 in AFSI are 1.0 and 0.10, compared to 0.8 and 0.55 for AISI respectively. Fig. 6 also exhibits that CESM and MIROC have similar characteristics, with substantial biases of t2m and mslp that are effectively removed by upper-level atmospheric nudging. Still in EC-Earth, for t2m, n.s.d. improves from 0.62 to 1.1 and mslp from 0.74 to 1.0. Thus, by nudging the upper-level atmosphere the surface variables get the correct magnitude. Fig. 6(c) also shows that the spatial

pattern correlation improves for prcp from 0.39 for the free running AISI run to 0.80 in the AFSI run. Another interesting observation obtained from comparing wave-5 and wave-7 Taylor Diagrams is that the models are more clustered for all variables for wave-7 compared to wave-5.

In general, nudging the soil moisture does not affect the upper atmospheric flow. AISF runs have similar biases in terms of

correlation, n.s.d. and RMSE, for t2m and prcp, compared to AISI, across the different models. The same conclusion stands for the AISF and AFSI runs for prcp and mslp. The aforementioned observations are location specific as one component within a climate model might, erroneously, be tuned in such a way that it compensates for biases in other components of the climate model. If so, nudging only that component would not reduce the overall bias. In this case, prescribing only the soil moisture part does not guarantee the reduction of the overall bias.


## 4 Discussion and outlook

### 4.1 Discussion

Large atmospheric circulation patterns, especially amplified wave-5 and wave-7 circumglobal Rossby waves, play an important role in climate variability and can trigger and maintain persistent extreme events such as heatwaves and prolonged precipitation periods during the summer months. In this study, we demonstrate that individual Fourier modes are well captured in different climate models in terms of their climatology and variability. The phase-locking behavior, in particular for wave-5 and wave-7, is captured. Both amplitude and week-to-week variability, in terms of standard deviations, are reasonably well reproduced in all models for all relevant wave numbers. The composites and bias metrics for the upper-level wave (v250) during wave-5 and wave-7 episodes show that the wave's amplitude and pattern are well-captured. Although the upper-level wind flows are satisfactorily reproduced across all models, their associated anomalies in surface variables (temperature, precipitation and sea level pressure) during high-amplitude wave-5 and wave-7 episodes are too weak. The MMM n.s.d.s for t2m and prcp are typically underestimated by a factor of 1.5 in wave-5 and wave-7 episodes. These model biases can be largely corrected by nudging the upper-level atmosphere. For instance, the n.s.d.s for the t2m, prcp, and mslp improve approximately by a factor in the range of 1.4 to 1.6 for wave-5 and wave-7 episodes. Such reductions in bias metrics are not observed when prescribing the soil moisture. This implies that when removing a nonsubstantial bias in the upper atmospheric levels (by nudging the circulation), this removes the relatively large biases in surface anomalies relevant for extremes. A full analysis of the underlying reasons is outside the scope of this paper, but here we discuss some potential mechanisms. First, nudging zonal (u) and meridional (v) winds in the upper-atmosphere, strongly constrains the large-scale vertical wind component ($\omega$) which is a key input for cloud parameterization schemes. In models, large-scale vertical wind is primarily defined by divergence in the horizontal wind fields, ensuring mass conservation, and thus nudging u and v will also effectively nudge $\omega$. Likewise, biases in u and v will propagate in $\omega$ and can then have a strong, possibly non-linear, impact on the amount of clouds in models (Satoh et al., ,2019; Rio et al., 2019). Regions with anomalously high pressure associated with a quasi-stationary Rossby wave, will have pronounced subsidence in ERA5 but this is likely too weak in the models. As this subsidence might not be well represented in the models, because of biases in the upper-level flow. As a consequence, the models might have hazier cloud conditions as compared to clear-sky conditions in ERA5. This would impact the surface by reduced short wave radiation and hence less pronounced warm anomalies. Potential limitations in the cloud parametrization schemes could exacerbate this, as models have difficulties reproducing clear sky conditions (Lacagnina and Selten 2014). The resolution for GCMs often do not allow sub-grid scale convective systems and their associated clouds to be resolved. More generally, particularly in mid-latitude continents, climatological biases in both clouds and precipitation persist in major GCMs (Rio et al., 2019). Specifically for EC-Earth, it has been shown that that there are too many clouds in in this model that are optically thick but too little clouds that are optically thin (Lacagnina and Selten 2014). Thus, this way, a relatively small bias in the upper-level horizontal wind fields could propagate via vertical wind and cloud scheme, into more substantial biases at the surface.

While previous work has indicated that soil moisture can have pronounced effects on quasi-stationary Rossby waves, including circumglobal waves (Koster et al., 2016; Teng et al., 2019), our analyses show that adjusting for soil moisture biases (by prescribing soil moisture) has little effect on the representation of circumglobal waves and their associated surface anomalies. These differences could arise from different time-scales and/or experiment set-ups. Earlier studies focused mainly on monthly to seasonal mean responses while we analyze weekly timescales. In addition, Teng et al. (2019) apply a very strong soil moisture anomly by setting it to zero over the western US, a region to which the climatological circumglobal wave might be particularly sensitive. In contrast, in our prescribed soil moisture experiments, the soil moisture is set to more realistic values, substantially larger than zero, coming from the model's land component forced by atmospheric fields from reanalysis. Our experiments thus represent much smaller forcings than those of Teng et al. (2019).

To be clear, our results are not questioning the importance of soil moisture as a prime driver of summer surface temperature extremes in various regions throughout the mid-latitudes. Rather, our study shows that prescribing the soil moisture in the models has little effect on surface variables and upper-level variables during high-amplitude wave episodes. Several studies have shown that soil moisture can play an important role in maintaining large-scale circulation anomalies associated with extremely warm and dry conditions (e.g. Erdenebat and Sato, 2018). In particular, under future climate change reduced soil moisture can lead to a higher probability of heatwaves in Europe during summer via interactions between the land surface and the atmosphere (e.g. Seneviratne et al., 2006). This, however, does not say anything about the imprint of soil moisture biases on biases in near-surface climate in relation to the Rossby wave events investigated in our study. Apparently, it is the state of the atmosphere, i.e. circulation or clouds and precipitation, that governs the model biases in near-surface climate and not so the state of the land surface. Further, in the interpretation of our results, one should be wary that in our prescribed soil moisture runs (AISF/AFSF), we prescribe the 'approximately observed' soil-moisture conditions (and not e.g. soil moisture climatology). This implies that the turbulent heat fluxes in AISF/AFSF still depend on this prescribed soil moisture condition. This means that during for example the heatwave period in Russia in 2010, the prescribed soil moisture will be anomalously dry, which will result in strong sensible heat fluxes.

## 4.2 Limitations and outlook

As with any choice of circulation metrics, our approach based on Fourier analyses of the zonally oriented wave component has its limitations. This approach implies that if a particular wavelength is pronounced in only one part of the hemisphere, this can result in a high-amplitude FFT signal. Thus, high-amplitude waves (as defined by our metrics) do not necessarily have to be circumglobal. They can either result from a circumglobal wave pattern or a pronounced regional wave pattern. Still, the fact that we find pronounced and significant wave-patterns in our composite analyses reveals that those reflect preferred wave positions. In particular, wave 5 and wave 7 are subject to this phase-locking behavior. Whenever those quasi-stationary waves

grow in amplitude, they tend to do so in the same longitudinal phase position thereby causing temperature and precipitation anomalies in the same geographical regions. This has been reported before for observational data, highlighting the risks this creates for multiple breadbasket failures (Kornhuber et al, 2020). The prime motivation of our study is to see how well climate models reproduce these waves, and to that end our FFT-based metric is useful. Further, the number of high-amplitude waves in ERA5 might be under-sampled which might lead to inconsistencies between models and ERA5. Since we use five ensemble

members for the AISI experiment, the amount of data is larger by a factor of five for the models compared to ERA5. As for the atmospheric nudging experiments, from the nudging vertical profile, it can be observed that in CESM and MIROC, the nudging intensity is identical, whereas in EC-Earth the nudging strength is weaker between 700hPa to 400hPa. However, we do not think that these differences between exact model setup and potential under-sampling in ERA-5 has a large effect on our main findings.

    Our findings have implications for climate model projections of persistent summer weather extremes in the key affected regions. Kornhuber et al. (2020) identified hotspots that are affected by summertime amplified wave-5 (Central North America, Eastern Europe, and Eastern Asia) and wave-7 (Western Central North America, Western Europe and Western Asia) patterns. These regions are sensitive to simultaneous heat extremes, and to exacerbate the situation, some identified regions are also

considered as global breadbasket regions. In summers when wave-5 or wave-7 events persist more than two weeks, the average reduction in crop production is 4% and even up to 11% on regional level (Kornhuber et al., 2020). In our study, for wave-5 and wave-7 events, all the aforementioned key regions are identified in our three models in terms of positive near-surface temperature anomalies. This gives more confidence in that those state-of-the-art climate models can be used to study present and future agricultural risks associated with such wave patterns. Since the strength of the near-surface temperature anomalies

is underestimated, the climate models are likely to underestimate heatwaves as well. A potential way to adjust for this is to establish statistical links between upper-level atmospheric flows to near-surface temperatures based on observational data. Then use this statistical link to adjust the effect of upper-level atmospheric circulation changes in climate models under future scenarios for heatwave risks. This approach is likely to be fruitful as our analyses suggest that upper-level waves are well represented across models. In addition, it will be important to assess how upper-level wave patterns change under future

greenhouse gas forcing, in terms of position, strength and persistence, and how that would effect surface extremes.

## 5 Summary and Conclusions

    Our validation study shows that upper-level wave characteristics are reasonably well reproduced in three GCMs in historic AMIP runs with MMM of n.s.d. for wave-5 and wave-7 episodes being 0.99 and 0.91. Both the climatology and phase-locking behaviors are captured in models for wave numbers 5 and 7, as the MMM correlation values are 0.90 and 0.83. Surface

temperature anomalies associated with amplified wave-5 and wave-7 patterns are weaker in models as compared to ERA5

reanalysis data. This bias in surface temperature anomalies during high amplitude wave-5 and wave-7 episodes is effectively removed when nudging the mid-to-upper level atmosphere.

In summary, for the meteorological variables analyzed in this study, we find that:

- Overall, v250 is the accurately represented and precipitation has the largest biases for both wave-5 and wave-7 episodes.
- Prescribing soil moisture does not improve the representation of surface anomalies in t2m and prcp.
- Nudging the upper-level atmosphere indicates that this is likely the prime origin of surface anomaly biases. We observe significant improvements from AISI and AISF runs to AFSI runs across all models and all variables.

We conclude that the relatively pronounced bias in the surface anomalies for amplified wave episodes, mainly originates from smaller biases in the upper-level atmospheric circulation. Our study suggests that climate models can be used to study present and future wave characteristics, but that care should be taken when analyzing the associated surface extremes.

*Code and data availability*. The code and data can be made available by the authors upon request.

*Author contributions.* FL, DC, and FS designed the analysis with input from KK. FL ran the EC-Earth3 simulations with technical help from FS, WM, PLS, and TR. HS ran the interactive and atmosphere nudged simulations with MIROC5. DT and HK ran the soil moisture nudged simulations with MIROC5. KW ran the CESM1.2 model simulations with technical support by Mathias Hauser. FL analyzed the results from all models. All authors contributed to the discussion of results. FL prepared the manuscript with contributions from all co-authors.

*Competing interests.* The authors declare that they have no conflict of interest.

*Acknowledgements.* EC-Earth3 simulations were contributed by VU Amsterdam and KNMI. The MIROC5 simulations were contributed by NIES Japan and University of Tokyo. CESM1.2 simulations were contributed by ETH Zurich. The authors thank editor Dr. Irina Rudeva and three anonymous reviewers for their constructive and insightful comments and suggestions to the manuscript. The authors would like to thank Mathias Hauser and Emanuel Dutra for their help in the discussions for the preparation of soil moisture prescription data and the applications in the models. FL, DC, and FS acknowledge VIDI-award

from Netherlands Organization for Scientific Research (NWO) (Persistent Summer Extremes "PERSIST" project: 016.Vidi.171.011). KK was partially supported by the NSF project NSF AGS-1934358.KW and SIS acknowledge funding from the European Research Council (ERC) ("DROUGHT-HEAT" project, Grant 617518). HS was supported by the 455 Integrated Research Program for Advancing Climate Models (JPMXD0717935457). The MIROC5 simulations were performed by using Earth Simulator in JAMSTEC and the NEC SX in NIES. WM is supported through the Swedish strategic research area ModElling the Regional and Global Earth system (MERGE). H.K acknowledges the National Research Foundation of Korea (NRF) grant Funded by the Korea Government (MSIT)(2021H1D3A2A03097768).

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

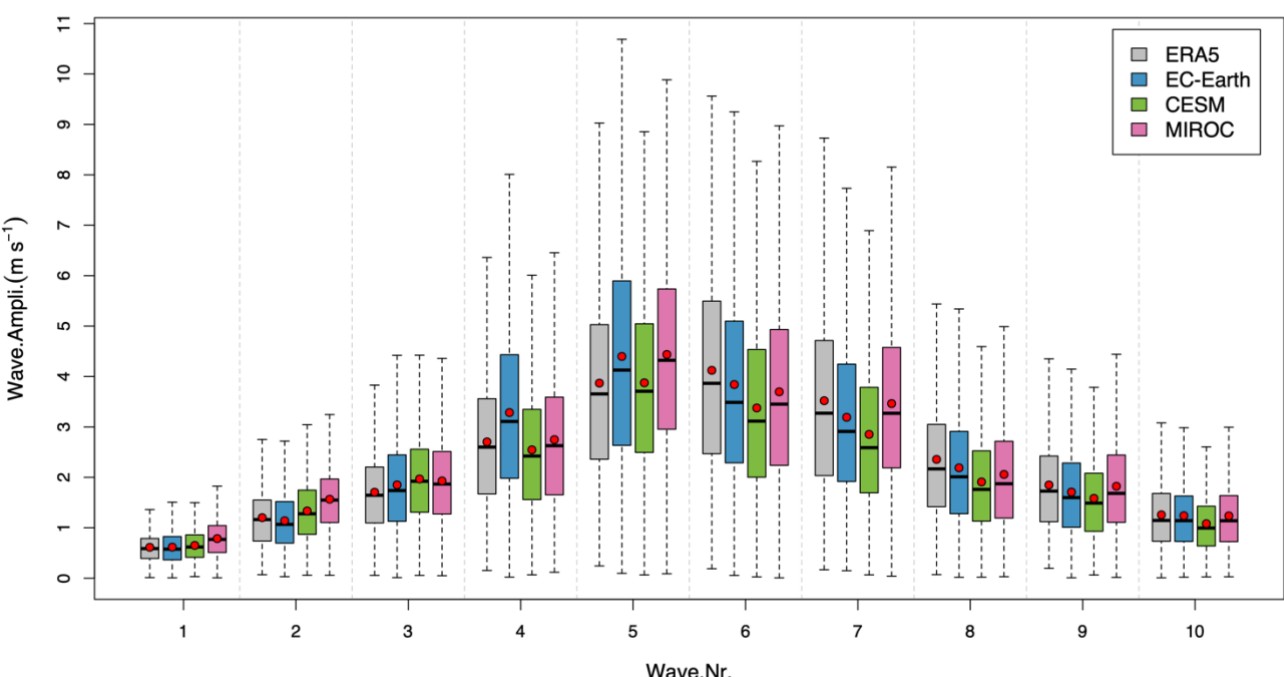

**Figure 1: Boxplot for wave amplitudes in AISI climatology runs for climate models EC-Earth, CESM, and MIROC, as well as reanalysis data ERA5 for the period of June, July, and August in 1979-2015/2016. Red dots indicate the mean, and thick black lines represent the median. The lower hinge of each box is Q1 quartile (25th), and the upper hinge for Q3 quartile (75th). The upper and lower bars are based on the 1.5 times interquartile range (IQR) value. The outliers are not shown in the plot.**


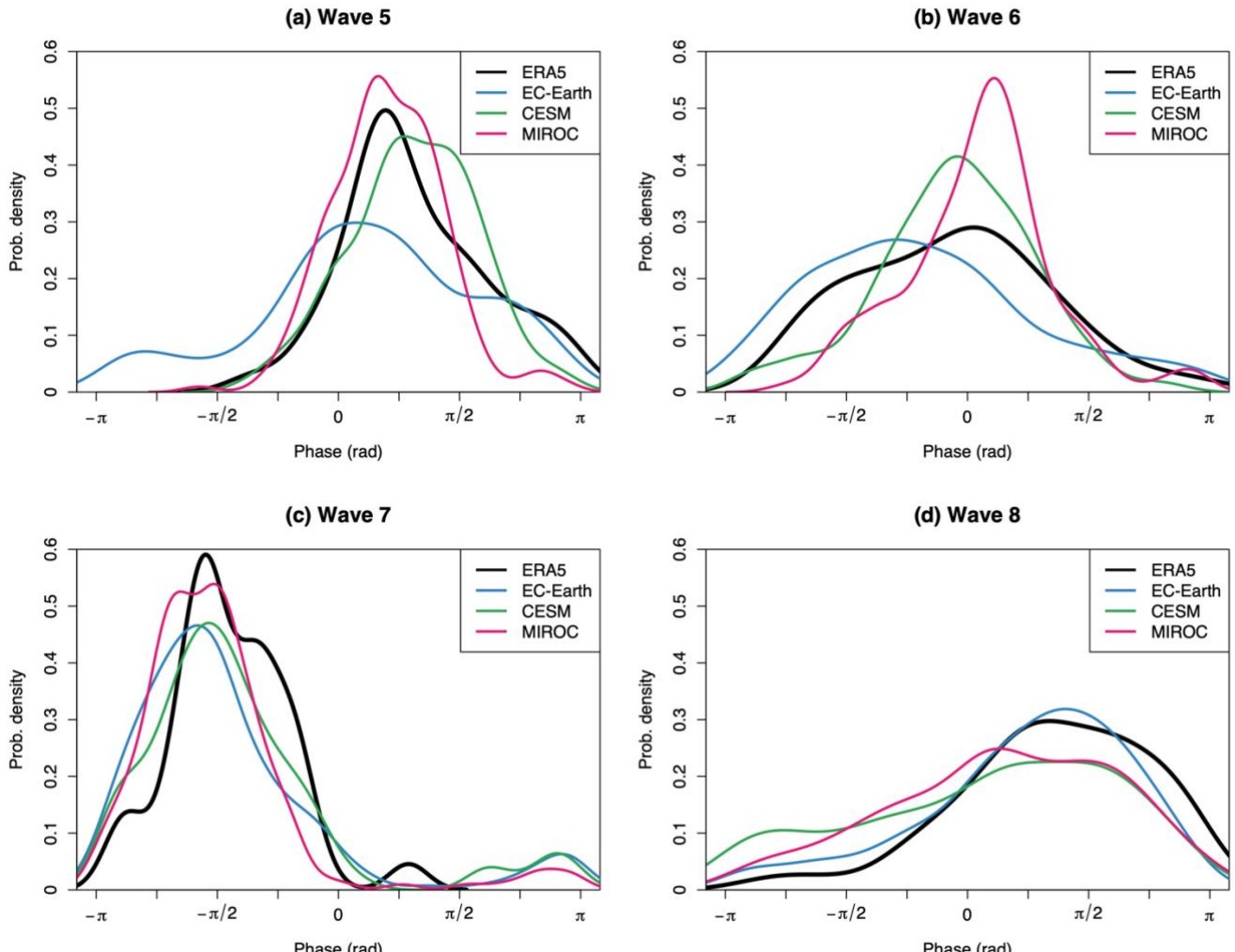

Figure 2: **Phase-locking of Rossby waves for JJA ERA5 and model waves 5-8 in control run AISI for high wave amplitude episodes (> 1.5 s.d.): (a)-(d), Probability density functions of the phase positions of waves 5-8 in ERA5, EC-Earth, CESM, and MIROC during JJA for the period of 1979-2015/2016 (wave 5 (a), wave 6 (b), wave 7 (c), wave 8 (d)). The bandwidth for ERA5 and models are as follows: (a) Wave 5: 0.35(ERA5), 0.40(EC-Earth), 0.25(CESM), 0.22(MIROC), (b) Wave 6: 0.53(ERA5), 0.45(EC-Earth), 0.30(CESM), 0.25(MIROC), (c) Wave 7: 0.25(ERA5), 0.29(EC-Earth), 0.27(CESM), 0.22(MIROC), (d) Wave 8: 0.49(ERA5), 0.39(EC-Earth), 0.52(CESM), 0.46(MIROC).**

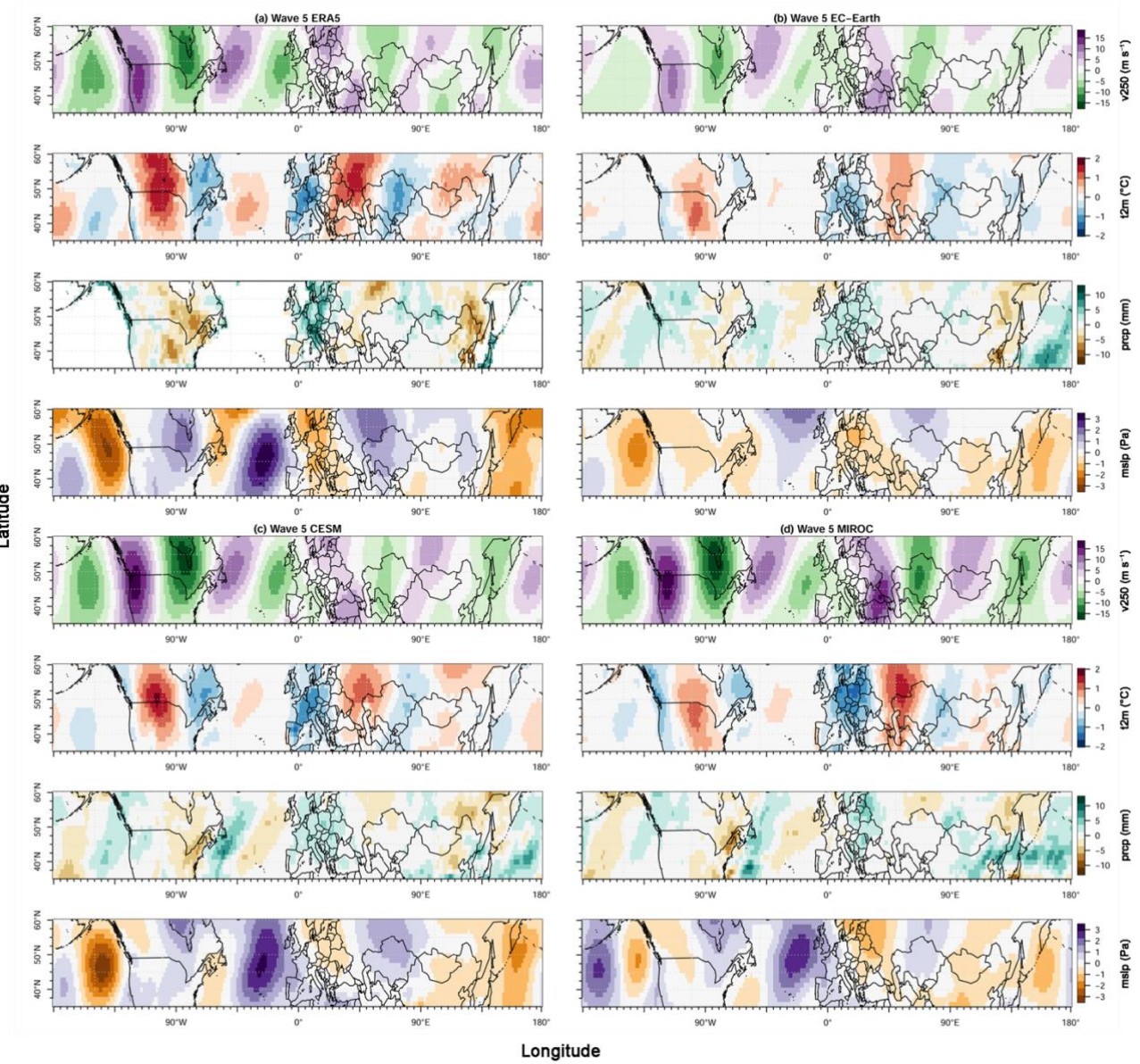

**Figure 3: Composite anomaly plots of weeks with high-amplitude waves-5 episodes for meridional wind velocity at 250hPa (v250, absolute field), near-surface temperature (t2m, anomaly), precipitation (prcp, anomaly), and sea level pressure (mslp, anomaly) in ERA5 (a), EC-Earth (b), CESM (c) and MIROC (d) based on control runs AISI.**


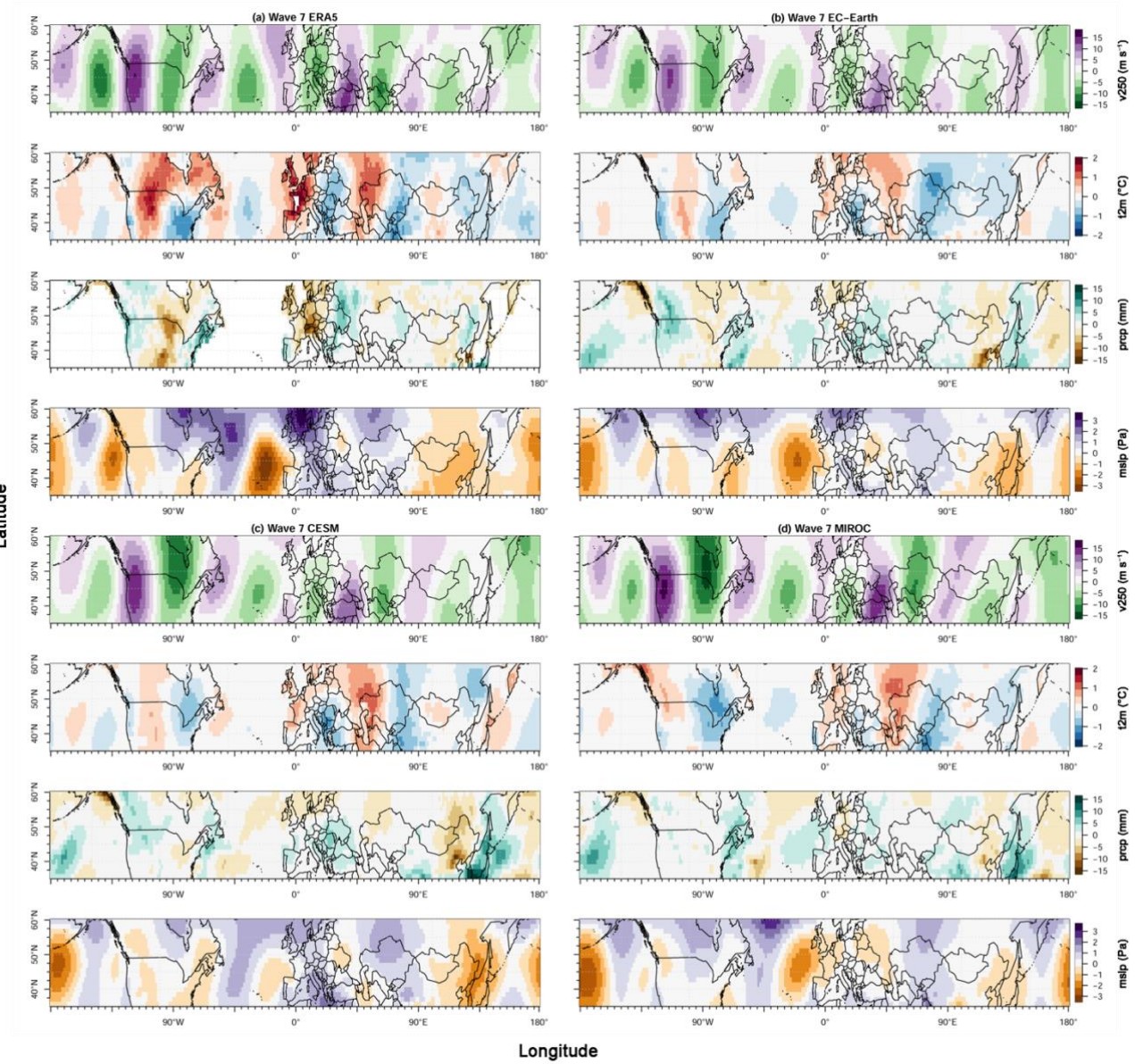

**Figure 4: Composite anomaly plots of weeks with high-amplitude waves-7 episodes for meridional wind velocity at 250hPa (v250, absolute field), near-surface temperature (t2m, anomaly), precipitation (prcp, anomaly), and sea level pressure (mslp, anomaly) in ERA5 (a), EC-Earth (b), CESM (c) and MIROC (d) based on control runs AISI.**


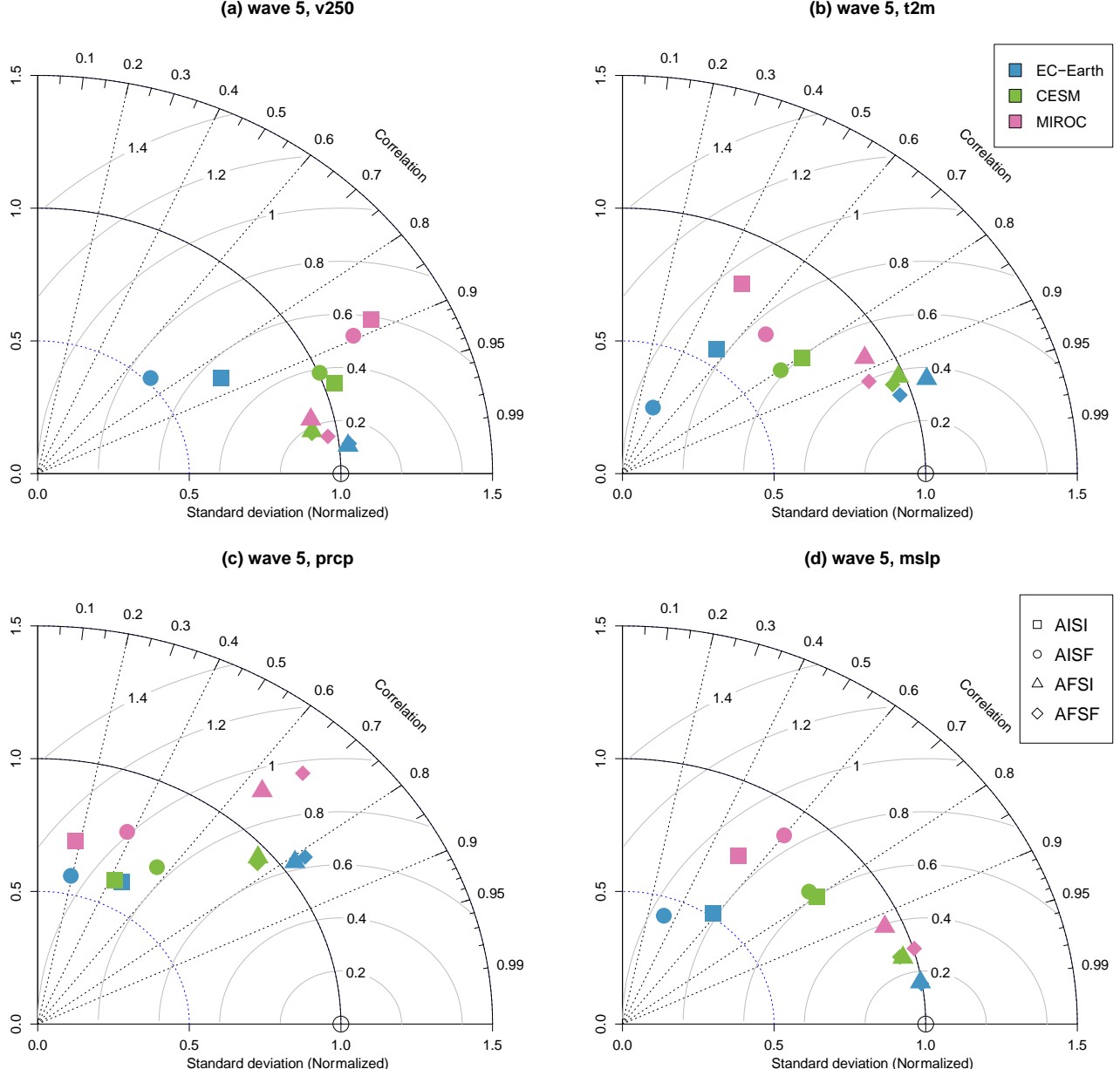

**Figure 5: Taylor Diagram for all experiments in models compared to ERA5 for wave-5 events and episodes. For a) v250, b) t2m, c) prcp, and d) mslp, the Taylor diagram presents for each model and each experiment, three statistics: the Pearson correlation (dashed lines); the RMS error (grey contours); and the normalized spatial standard deviation (solid black contours).**

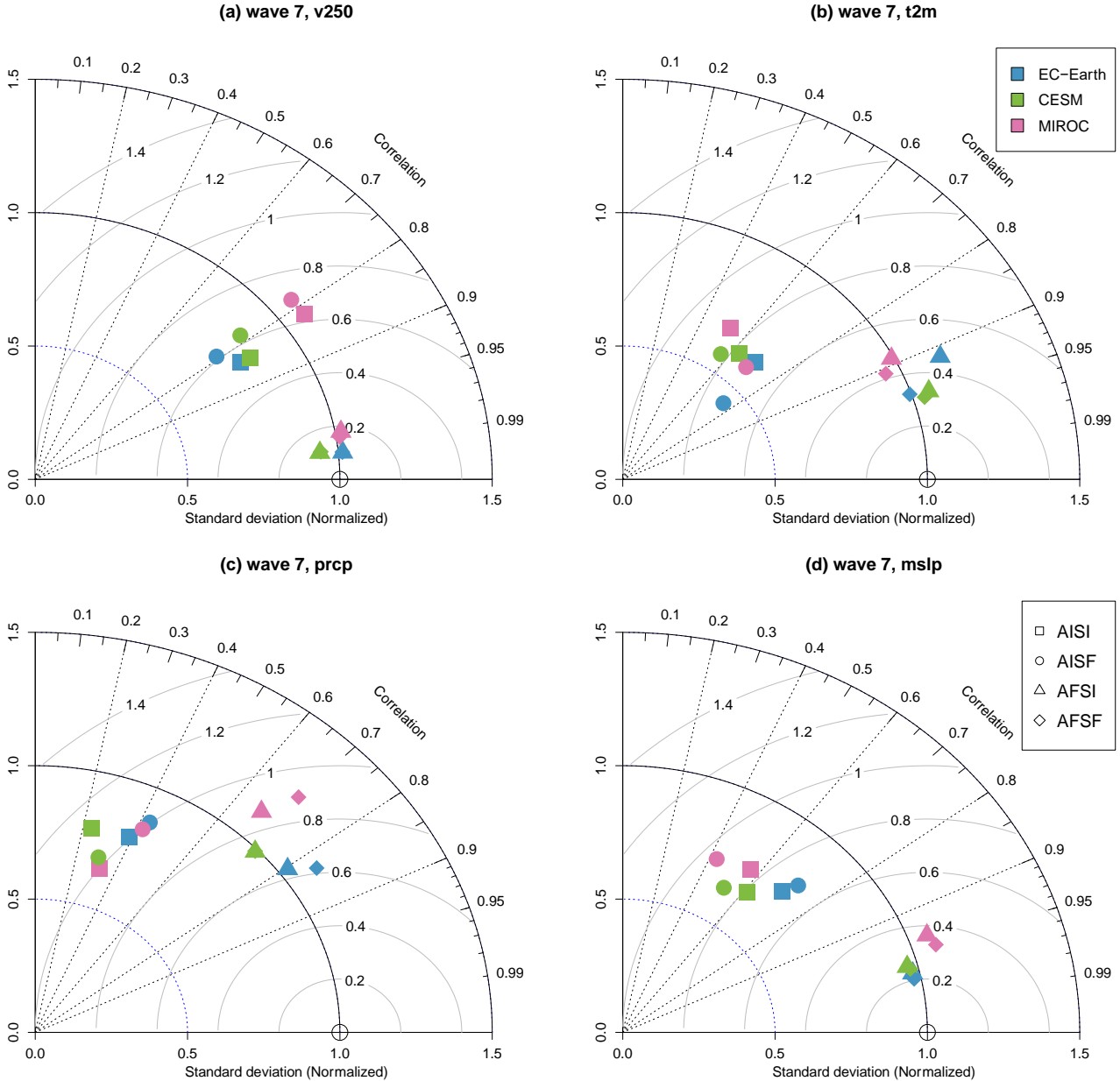

**Figure 6: Taylor Diagram for all experiments in models compared to ERA5 for wave-7 events and episodes. For a) v250, b) t2m, c) prcp, and d) mslp, the Taylor diagram presents for each model and each experiment, three statistics: the Pearson correlation (dashed lines); the RMS error (grey contours); and the normalized spatial standard deviation (solid black contours).**

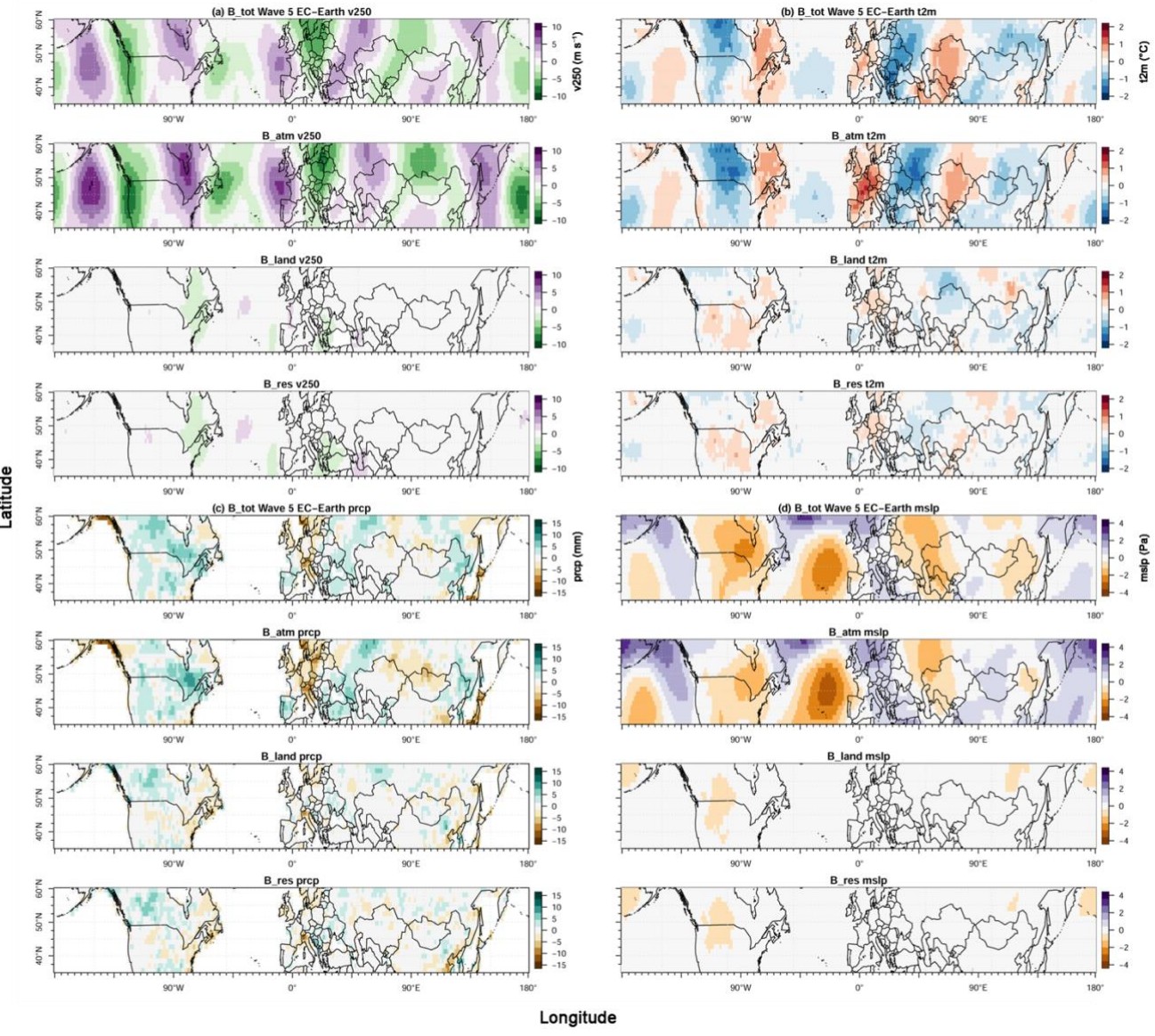


**Figure 7: Bias plots for high-amplitude wave-5 events and episodes in different experiments for EC-Earth. Total bias(B_tot), atmospheric bias(B_atm), land-atmosphere interaction bias (B_land) and residual bias(B_res) for meridional wind velocity at 250hPa (a), surface temperature (b), precipitation (c), and sea level pressure (d).**

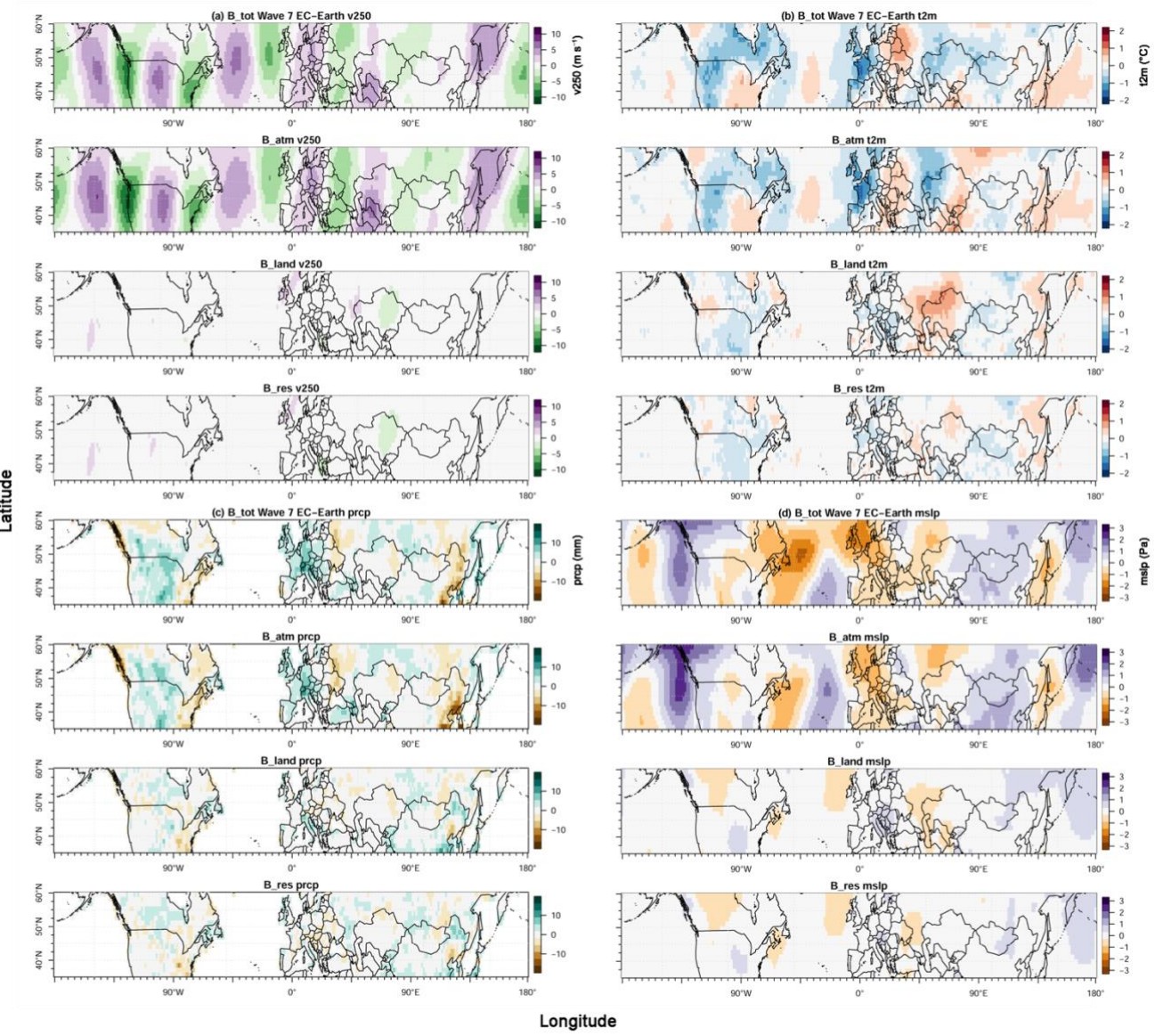


**Figure 8: Bias plots for high-amplitude wave-7 events and episodes in different experiments for EC-Earth. Total bias(B_tot), Atmospheric bias(B_atm), Land-Atm interaction bias(B_land) and residual bias(B_res) for meridional wind velocity at 250hPa (a), surface temperature (b), precipitation (c), and sea level pressure (d).**


# Appendix

| Wave5 | v250 | | | | t2m | | | | prcp | | | | mslp | | | |
|---|---|---|---|---|---|---|---|---|---|---|---|---|---|---|---|---|
| Model | EC-Earth | CESM | MIROC | Model Mean | EC-Earth | CESM | MIROC | Model Mean | EC-Earth | CESM | MIROC | Model Mean | EC-Earth | CESM | MIROC | Model Mean |
| AISI | 0,858 | 0,945 | 0,884 | 0,896 | 0,552 | 0,806 | 0,481 | 0,613 | 0,460 | 0,427 | 0,179 | 0,355 | 0,584 | 0,801 | 0,515 | 0,633 |
| AISF | 0,719 | 0,926 | 0,895 | 0,847 | 0,375 | 0,801 | 0,669 | 0,615 | 0,192 | 0,554 | 0,377 | 0,374 | 0,317 | 0,777 | 0,600 | 0,565 |
| AFSI | 0,995 | 0,985 | 0,975 | 0,985 | 0,942 | 0,928 | 0,878 | 0,916 | 0,812 | 0,756 | 0,645 | 0,738 | 0,987 | 0,965 | 0,921 | 0,958 |
| AFSF | 0,994 | 0,986 | 0,989 | 0,990 | 0,951 | 0,936 | 0,920 | 0,936 | 0,814 | 0,768 | 0,679 | 0,754 | 0,988 | 0,964 | 0,959 | 0,970 |
| Wave7 | v250 | | | | t2m | | | | prcp | | | | mslp | | | |
| Model | EC-Earth | CESM | MIROC | Model Mean | EC-Earth | CESM | MIROC | Model Mean | EC-Earth | CESM | MIROC | Model Mean | EC-Earth | CESM | MIROC | Model Mean |
| AISI | 0,837 | 0,839 | 0,819 | 0,832 | 0,703 | 0,629 | 0,528 | 0,620 | 0,388 | 0,236 | 0,323 | 0,316 | 0,704 | 0,613 | 0,566 | 0,628 |
| AISF | 0,791 | 0,780 | 0,780 | 0,784 | 0,757 | 0,565 | 0,693 | 0,672 | 0,431 | 0,301 | 0,420 | 0,384 | 0,723 | 0,522 | 0,429 | 0,558 |
| AFSI | 0,995 | 0,994 | 0,984 | 0,991 | 0,915 | 0,950 | 0,890 | 0,918 | 0,804 | 0,729 | 0,668 | 0,734 | 0,974 | 0,967 | 0,939 | 0,960 |
| AFSF | 0,995 | 0,994 | 0,988 | 0,992 | 0,947 | 0,955 | 0,909 | 0,937 | 0,832 | 0,728 | 0,700 | 0,753 | 0,979 | 0,970 | 0,952 | 0,967 |

**Table A1: Summary of Model Taylor Diagram correlation values.**

| Wave5 | v250 | | | | | t2m | | | | | prcp | | | | | mslp | | | | |
|---|---|---|---|---|---|---|---|---|---|---|---|---|---|---|---|---|---|---|---|---|
| Model | EC-Earth | CESM | MIROC | Model Mean | NORM | EC-Earth | CESM | MIROC | Model Mean | NORM | EC-Earth | CESM | MIROC | Model Mean | NORM | EC-Earth | CESM | MIROC | Model Mean | NORM |
| AISI | 3,510 | 5,168 | 6,203 | 4,960 | 0,994 | 0,292 | 0,382 | 0,424 | 0,366 | 0,705 | 1,746 | 1,729 | 2,027 | 1,834 | 0,634 | 0,586 | 0,916 | 0,847 | 0,783 | 0,685 |
| AISF | 2,584 | 5,012 | 5,805 | 4,467 | 0,895 | 0,139 | 0,338 | 0,367 | 0,281 | 0,542 | 1,646 | 2,054 | 2,262 | 1,987 | 0,687 | 0,492 | 0,906 | 1,016 | 0,805 | 0,704 |
| AFSI | 5,133 | 4,584 | 4,608 | 4,775 | 0,957 | 0,553 | 0,509 | 0,472 | 0,511 | 0,985 | 3,024 | 2,779 | 3,322 | 3,042 | 1,052 | 1,138 | 1,096 | 1,075 | 1,103 | 0,965 |
| AFSF | 5,158 | 4,576 | 4,826 | 4,853 | 0,973 | 0,499 | 0,494 | 0,459 | 0,484 | 0,933 | 3,135 | 2,731 | 3,723 | 3,196 | 1,105 | 1,141 | 1,085 | 1,148 | 1,125 | 0,983 |
| Wave7 | v250 | | | | | t2m | | | | | prcp | | | | | mslp | | | | |
| Model | EC-Earth | CESM | MIROC | Model Mean | NORM | EC-Earth | CESM | MIROC | Model Mean | NORM | EC-Earth | CESM | MIROC | Model Mean | NORM | EC-Earth | CESM | MIROC | Model Mean | NORM |
| AISI | 4,162 | 4,344 | 5,585 | 4,697 | 0,907 | 0,310 | 0,304 | 0,334 | 0,316 | 0,631 | 2,526 | 2,510 | 2,064 | 2,366 | 0,744 | 0,818 | 0,734 | 0,818 | 0,790 | 0,717 |
| AISF | 3,894 | 4,467 | 5,574 | 4,645 | 0,897 | 0,219 | 0,285 | 0,292 | 0,265 | 0,530 | 2,775 | 2,190 | 2,669 | 2,545 | 0,800 | 0,877 | 0,700 | 0,792 | 0,790 | 0,717 |
| AFSI | 5,250 | 4,864 | 5,276 | 5,130 | 0,991 | 0,570 | 0,530 | 0,496 | 0,532 | 1,063 | 3,278 | 3,150 | 3,537 | 3,322 | 1,044 | 1,076 | 1,063 | 1,170 | 1,103 | 1,002 |
| AFSF | 5,245 | 4,883 | 5,240 | 5,123 | 0,990 | 0,498 | 0,520 | 0,476 | 0,498 | 0,994 | 3,532 | 3,154 | 3,928 | 3,538 | 1,112 | 1,076 | 1,072 | 1,188 | 1,112 | 1,010 |

**Table A2: Summary of model standard deviation values.**

| Wave5 | v250 | | | | | t2m | | | | | prcp | | | | | mslp | | | | |
|---|---|---|---|---|---|---|---|---|---|---|---|---|---|---|---|---|---|---|---|---|
| Model | EC-Earth | CESM | MIROC | Model Mean | NORM | EC-Earth | CESM | MIROC | Model Mean | NORM | EC-Earth | CESM | MIROC | Model Mean | NORM | EC-Earth | CESM | MIROC | Model Mean | NORM |
| AISI | 2,675 | 1,695 | 2,942 | 2,437 | 0,488 | 0,446 | 0,319 | 0,498 | 0,421 | 0,811 | 2,302 | 2,425 | 2,924 | 2,550 | 0,882 | 0,956 | 0,687 | 1,025 | 0,889 | 0,778 |
| AISF | 3,610 | 1,927 | 2,600 | 2,712 | 0,544 | 0,497 | 0,343 | 0,397 | 0,412 | 0,794 | 2,671 | 2,251 | 2,530 | 2,484 | 0,859 | 1,093 | 0,724 | 0,995 | 0,937 | 0,820 |
| AFSI | 0,542 | 0,944 | 1,140 | 0,875 | 0,175 | 0,187 | 0,209 | 0,257 | 0,218 | 0,420 | 1,771 | 1,822 | 2,490 | 2,027 | 0,701 | 0,189 | 0,303 | 0,450 | 0,314 | 0,274 |
| AFSF | 0,586 | 0,908 | 0,740 | 0,745 | 0,149 | 0,160 | 0,197 | 0,212 | 0,190 | 0,366 | 1,793 | 1,767 | 2,468 | 2,009 | 0,695 | 0,187 | 0,305 | 0,330 | 0,274 | 0,240 |
| Wave7 | v250 | | | | | t2m | | | | | prcp | | | | | mslp | | | | |
| Model | EC-Earth | CESM | MIROC | Model Mean | NORM | EC-Earth | CESM | MIROC | Model Mean | NORM | EC-Earth | CESM | MIROC | Model Mean | NORM | EC-Earth | CESM | MIROC | Model Mean | NORM |
| AISI | 2,838 | 2,813 | 3,263 | 2,971 | 0,574 | 0,380 | 0,397 | 0,442 | 0,407 | 0,812 | 3,076 | 3,395 | 2,988 | 3,153 | 0,991 | 0,785 | 0,872 | 0,928 | 0,862 | 0,783 |
| AISF | 3,174 | 3,267 | 3,582 | 3,341 | 0,645 | 0,376 | 0,420 | 0,372 | 0,389 | 0,777 | 3,062 | 3,152 | 2,878 | 3,031 | 0,953 | 0,785 | 0,949 | 1,04764 | 0,927 | 0,842 |
| AFSI | 0,527 | 0,638 | 0,935 | 0,700 | 0,135 | 0,231 | 0,167 | 0,237 | 0,212 | 0,423 | 1,988 | 2,262 | 2,558 | 2,269 | 0,713 | 0,250 | 0,306 | 0,510 | 0,355 | 0,323 |
| AFSF | 0,536 | 0,645 | 0,829 | 0,670 | 0,129 | 0,166 | 0,155 | 0,215 | 0,179 | 0,356 | 1,908 | 2,206 | 2,560 | 2,224 | 0,699 | 0,226 | 0,304 | 0,483 | 0,338 | 0,307 |

**Table A3: Summary of model RMSE values.**

| Wave5 | v250 | | | t2m | | | prcp | | | mslp | | |
|---|---|---|---|---|---|---|---|---|---|---|---|---|
| Model | corr | std | rmse | corr | std | rmse | corr | std | rmse | corr | std | rmse |
| AISI | 0,896 | 0,994 | 0,488 | 0,613 | 0,705 | 0,811 | 0,355 | 0,634 | 0,882 | 0,633 | 0,685 | 0,778 |
| AISF | 0,847 | 0,895 | 0,544 | 0,615 | 0,542 | 0,794 | 0,374 | 0,687 | 0,859 | 0,565 | 0,704 | 0,820 |
| AFSI | 0,985 | 0,957 | 0,175 | 0,916 | 0,985 | 0,420 | 0,738 | 1,052 | 0,701 | 0,958 | 0,965 | 0,274 |
| AFSF | 0,990 | 0,973 | 0,149 | 0,936 | 0,933 | 0,366 | 0,754 | 1,105 | 0,695 | 0,970 | 0,983 | 0,240 |
| **Wave7** | **v250** | | | **t2m** | | | **prcp** | | | **mslp** | | |
| Model | corr | std | rmse | corr | std | rmse | corr | std | rmse | corr | std | rmse |
| AISI | 0,832 | 0,907 | 0,574 | 0,620 | 0,631 | 0,812 | 0,316 | 0,744 | 0,991 | 0,628 | 0,717 | 0,783 |
| AISF | 0,784 | 0,897 | 0,645 | 0,672 | 0,530 | 0,777 | 0,384 | 0,800 | 0,953 | 0,558 | 0,717 | 0,842 |
| AFSI | 0,991 | 0,991 | 0,135 | 0,918 | 1,063 | 0,423 | 0,734 | 1,044 | 0,713 | 0,960 | 1,002 | 0,323 |
| AFSF | 0,992 | 0,990 | 0,129 | 0,937 | 0,994 | 0,356 | 0,753 | 1,112 | 0,699 | 0,967 | 1,010 | 0,307 |

**Table A4: Summary of Multi-Model Mean Taylor Diagram values.**

| ERA5 s.d. | v250 | t2m | prcp | mslp |
|---|---|---|---|---|
| **wave 5** | 4,990 | 0,519 | 2,893 | 1,144 |
| **wave 7** | 5,177 | 0,501 | 3,181 | 1,101 |

**Table A5: Summary of ERA5 standard deviation values.**


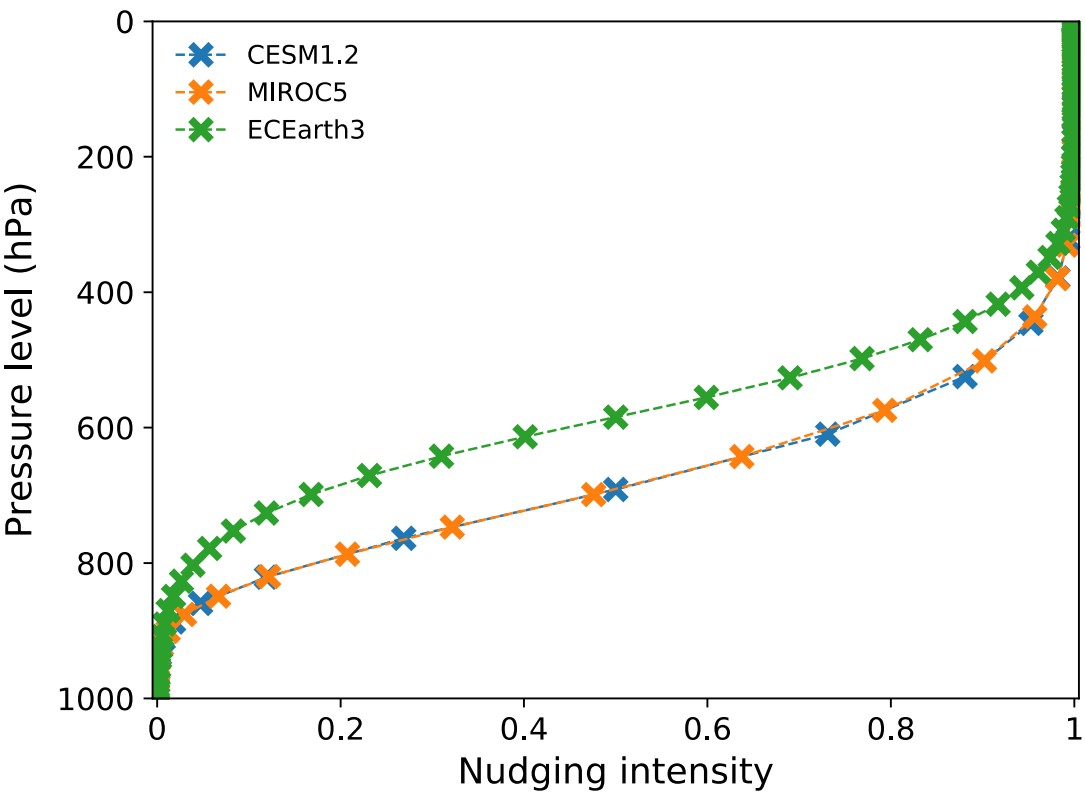

**Figure B1. Nudging profile for the three ExtremeX ESMs. The actual pressure levels are marked with an x and joined with lines.**
**The nudging intensity is given from zero (no nudging) to one (fully nudged).**



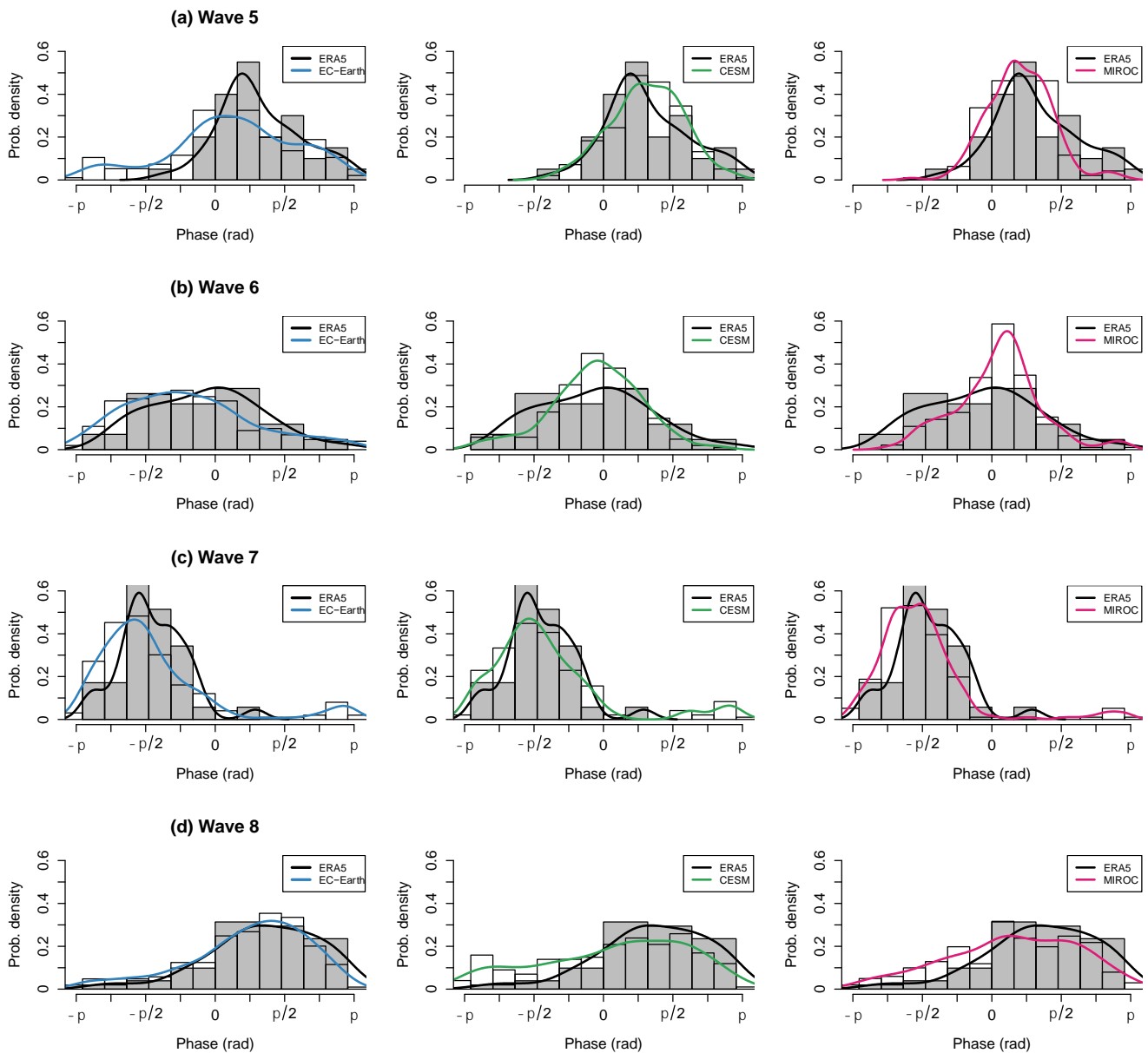

**Figure B2. Same as Fig.2 but with histogram added.**

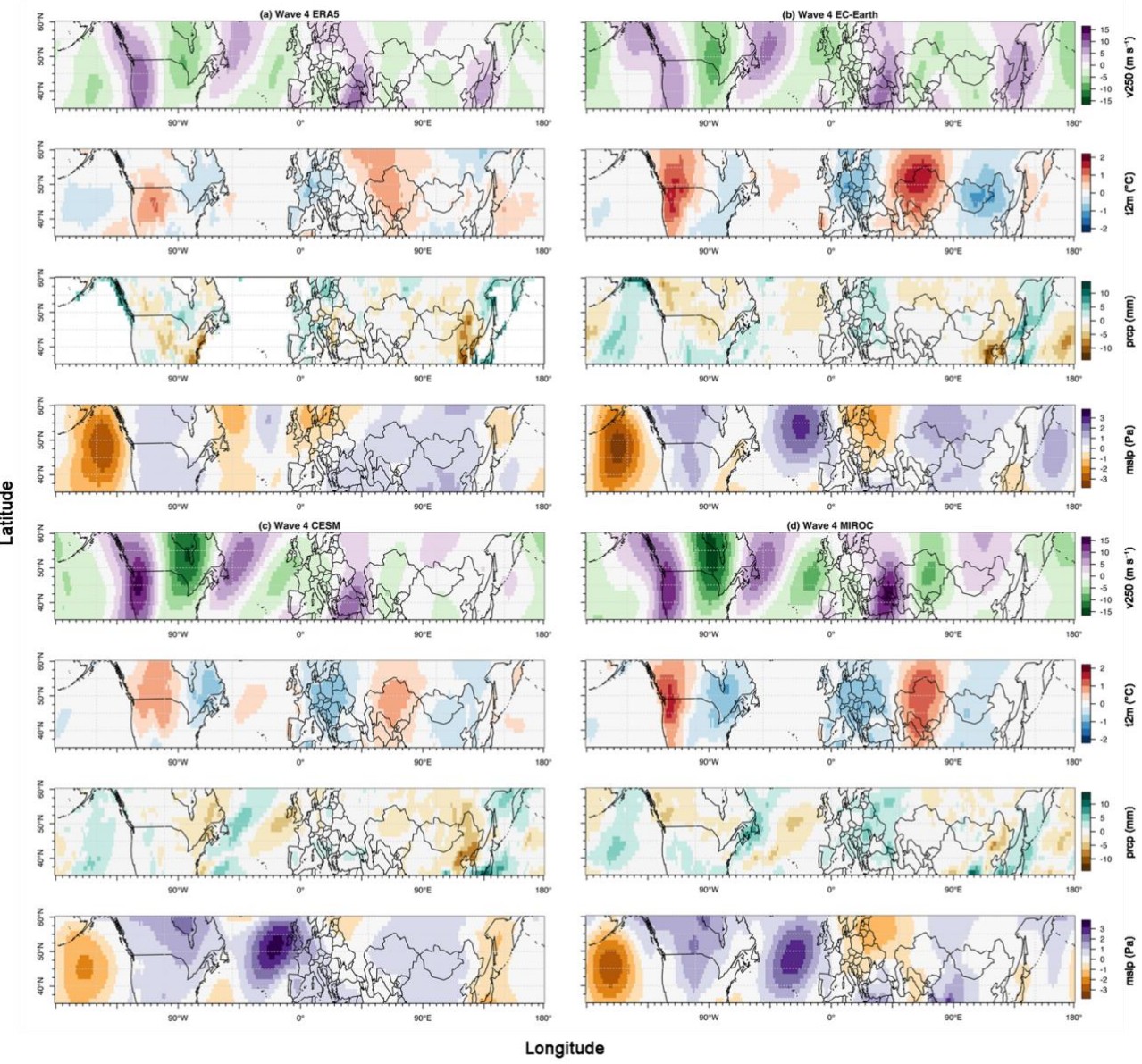

Figure B3: Composite anomaly plots of weeks with high-amplitude wave-4 episodes for meridional wind velocity at 250hPa (v250, absolute field), near-surface temperature (t2m, anomaly), precipitation (prcp, anomaly), and sea level pressure (mslp, anomaly) in ERA5 (a), EC-Earth (b), CESM (c) and MIROC (d) based on control runs AISI.

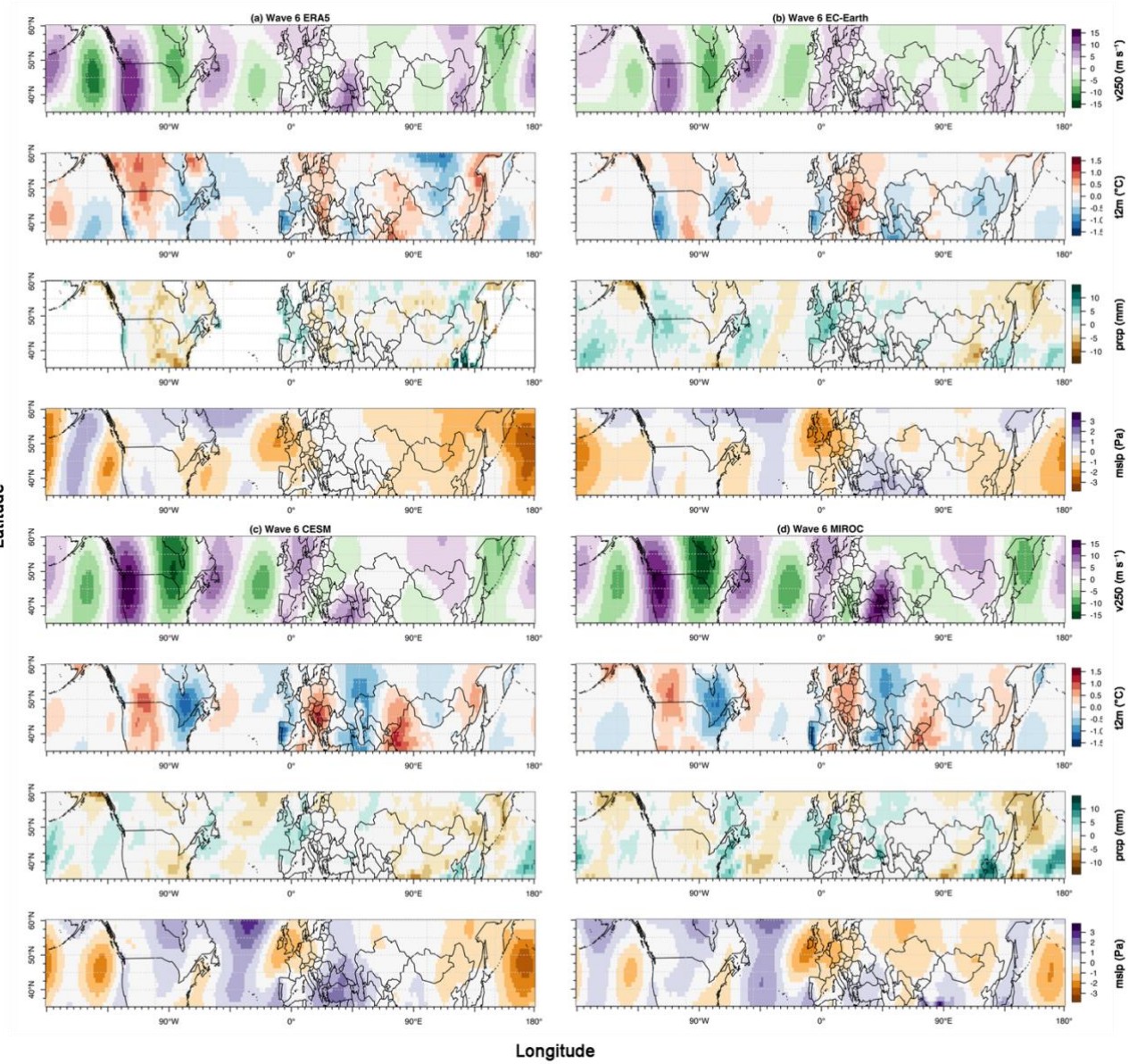


**Figure B4: Composite anomaly plots of weeks with high-amplitude wave-6 episodes for meridional wind velocity at 250hPa (v250, absolute field), near-surface temperature (t2m, anomaly), precipitation (prcp, anomaly), and sea level pressure (mslp, anomaly) in ERA5 (a), EC-Earth (b), CESM (c) and MIROC (d) based on control runs AISI.**

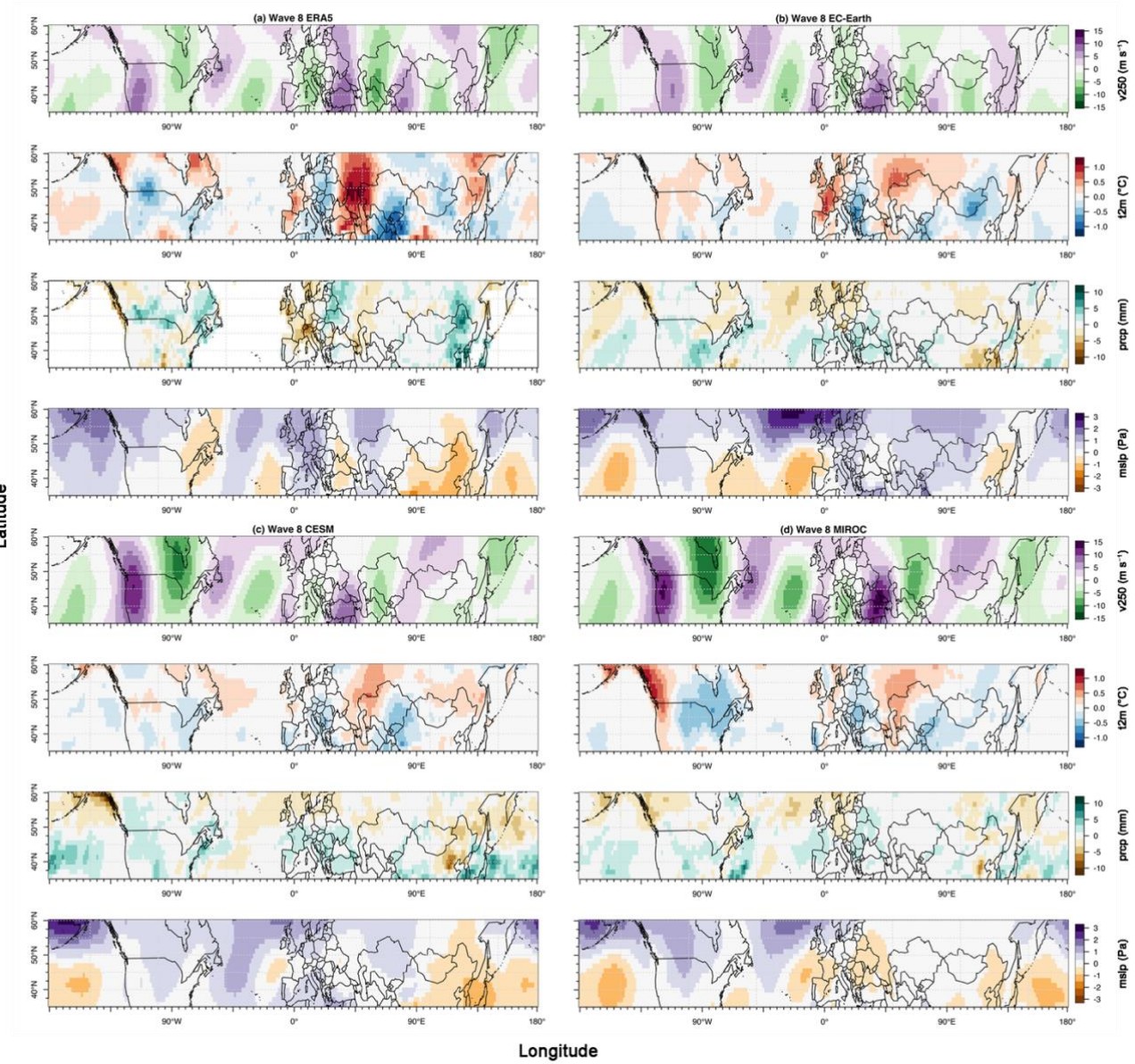

**Figure B5: Composite anomaly plots of weeks with high-amplitude wave-8 episodes for meridional wind velocity at 250hPa (v250, absolute field), near-surface temperature (t2m, anomaly), precipitation (prcp, anomaly), and sea level pressure (mslp, anomaly) in ERA5 (a), EC-Earth (b), CESM (c) and MIROC (d) based on control runs AISI.**

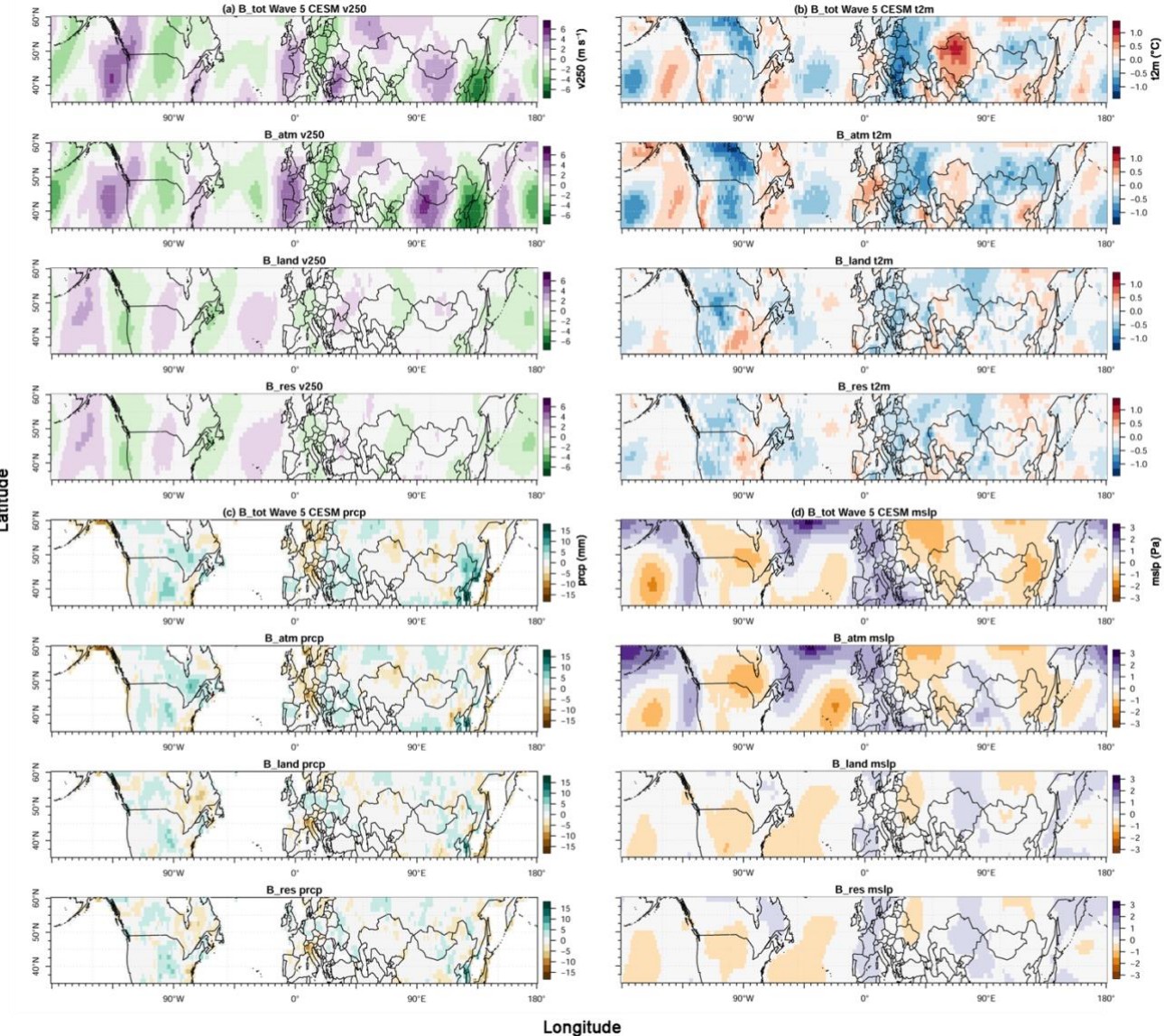


**Figure B6: Bias plots for high-amplitude wave-5 events and episodes in different experiments for CESM. Total bias(B_tot), atmospheric bias(B_atm), land-atmosphere interaction bias (B_land) and residual bias(B_res) for meridional wind velocity at 250hPa (a), surface temperature (b), precipitation (c), and sea level pressure (d).**


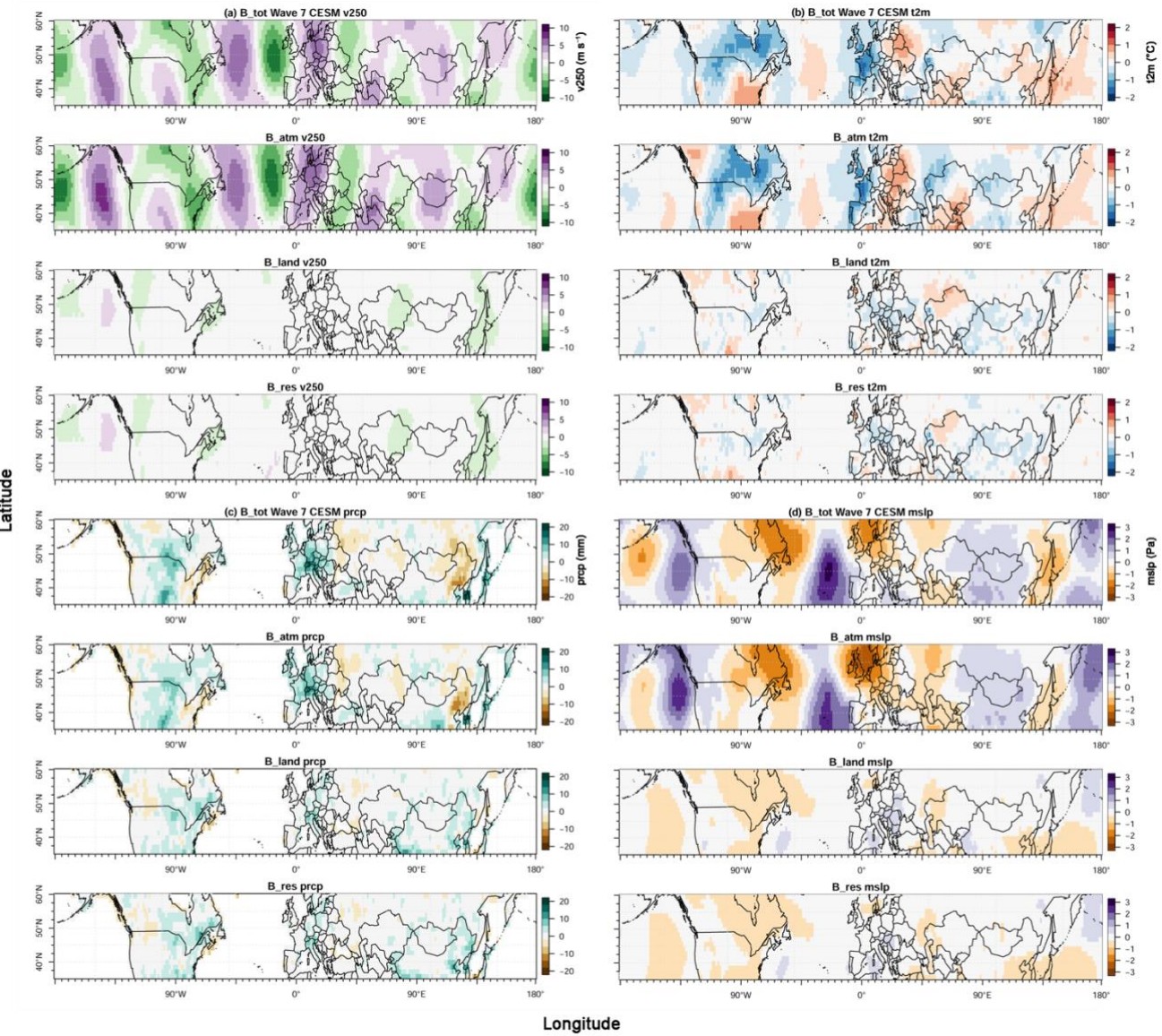

**Figure B7: Bias plots for high-amplitude wave-7 events and episodes in different experiments for CESM. Total bias(B_tot), Atmospheric bias(B_atm), Land-Atm interaction bias(B_land) and residual bias(B_res) for meridional wind velocity at 250hPa (a), surface temperature (b), precipitation (c), and sea level pressure (d).**

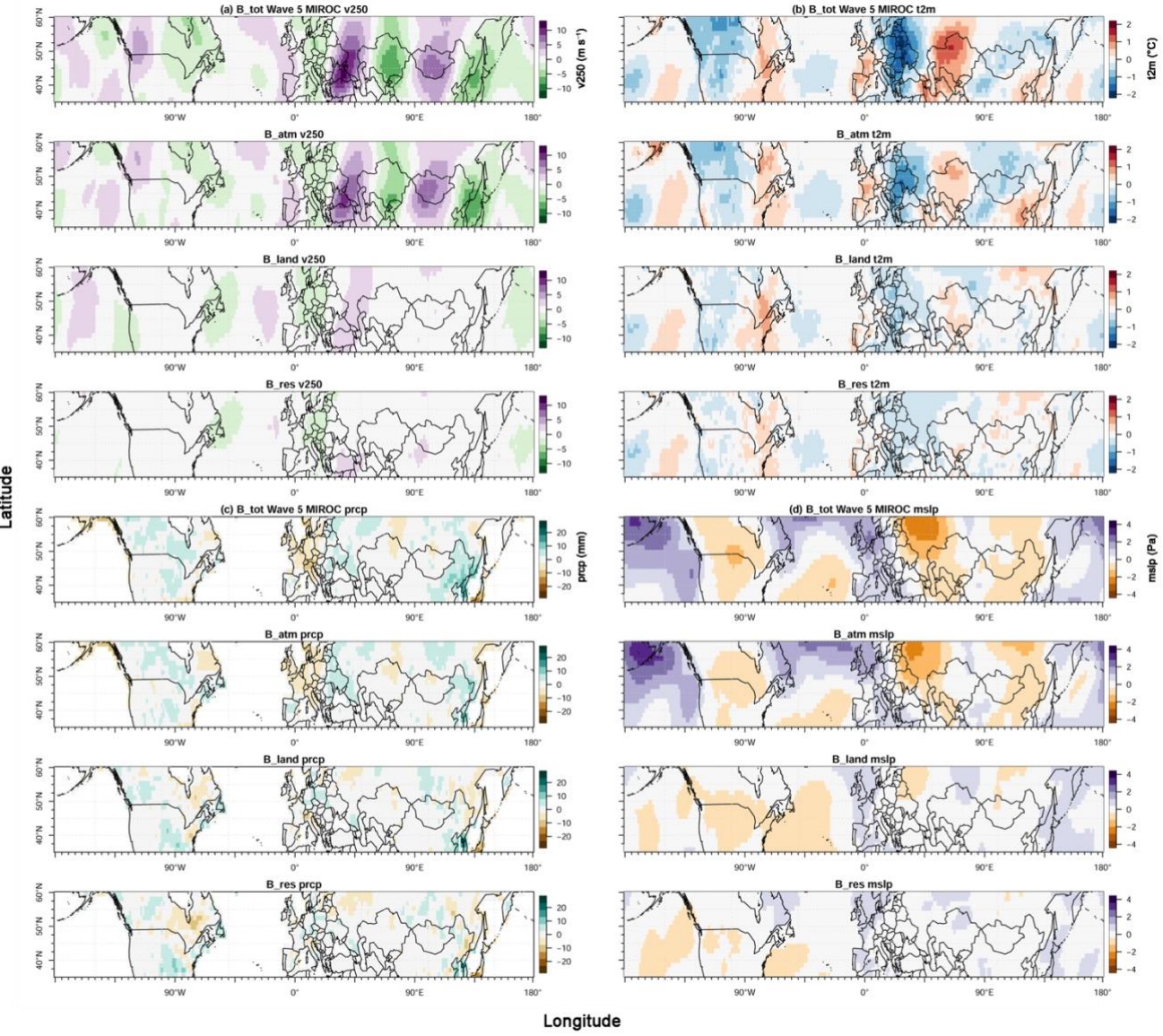

**Figure B8: Bias plots for high-amplitude wave-5 events and episodes in different experiments for MIROC. Total bias(B_tot), atmospheric bias(B_atm), land-atmosphere interaction bias (B_land) and residual bias(B_res) for meridional wind velocity at 250hPa (a), surface temperature (b), precipitation (c), and sea level pressure (d).**


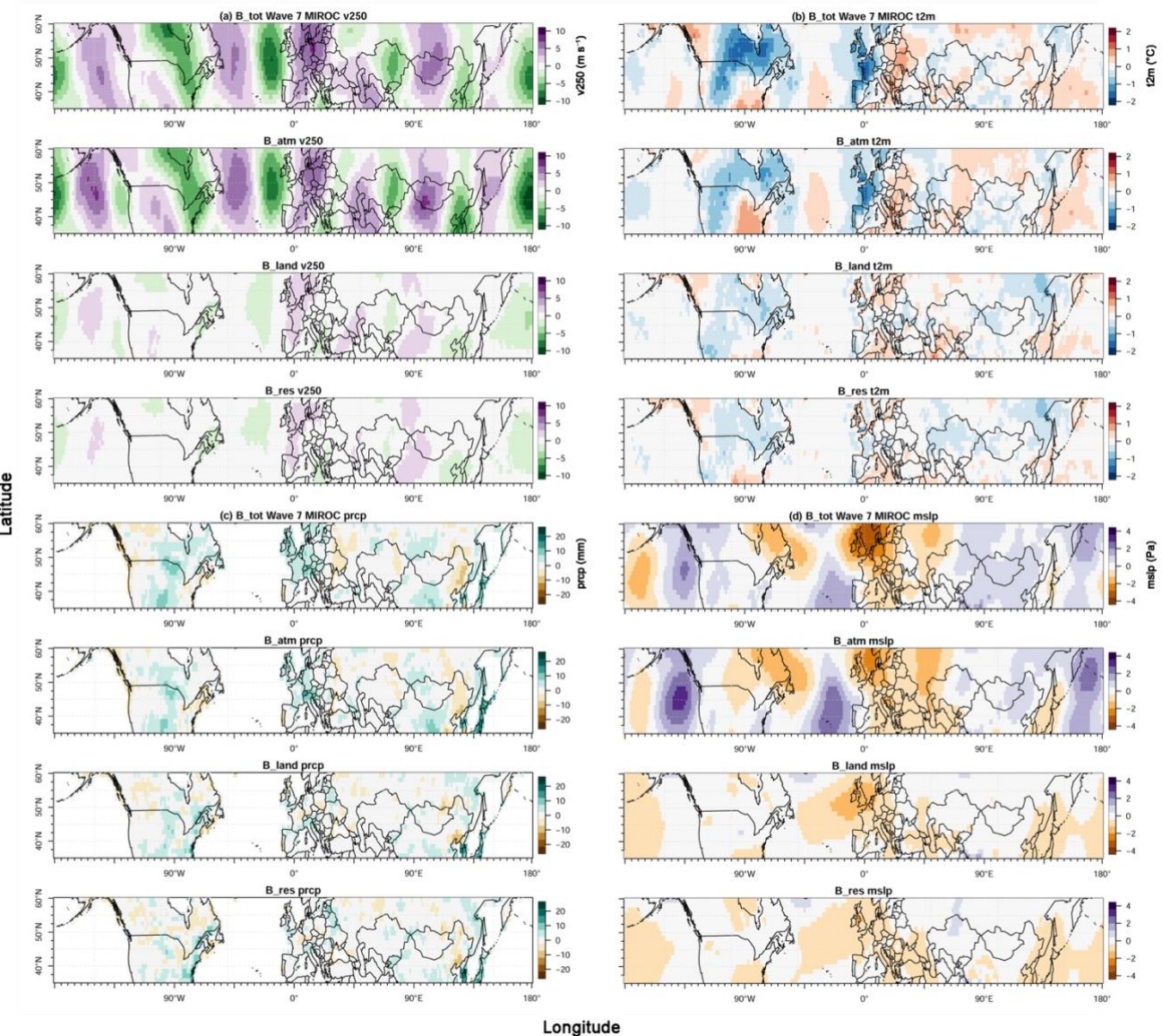

**Figure B9: Bias plots for high-amplitude wave-7 events and episodes in different experiments for MIROC. Total bias(B_tot), Atmospheric bias(B_atm), Land-Atm interaction bias(B_land) and residual bias(B_res) for meridional wind velocity at 250hPa (a), surface temperature (b), precipitation (c), and sea level pressure (d).**



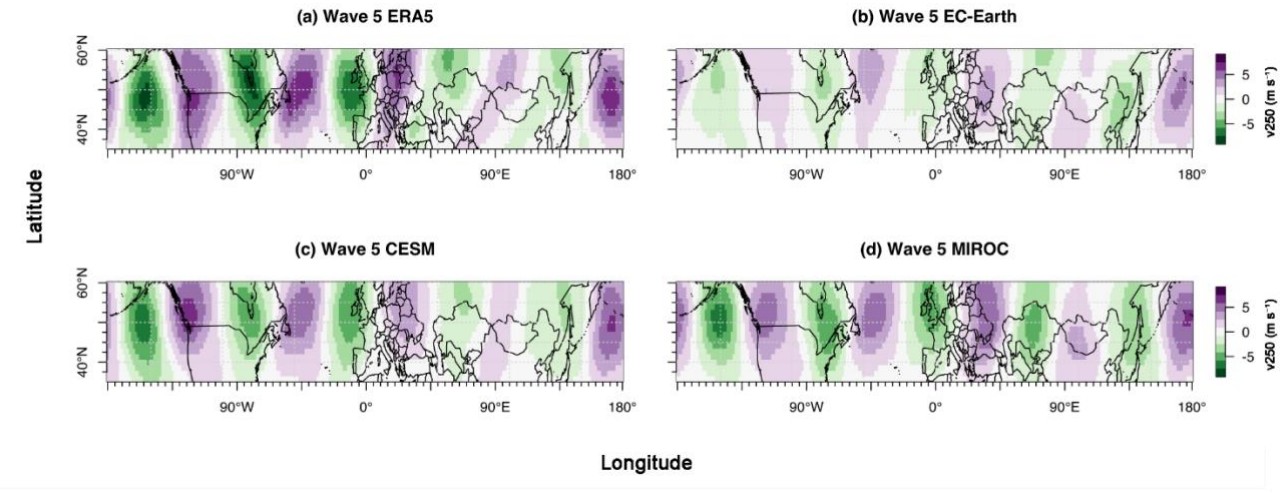

**Figure B10: Composite anomaly plots of weeks with high-amplitude wave-5 episodes for meridional wind velocity at 250hPa (v250, anomaly) in ERA5 (a), EC-Earth (b), CESM (c) and MIROC (d) based on control runs AISI.**

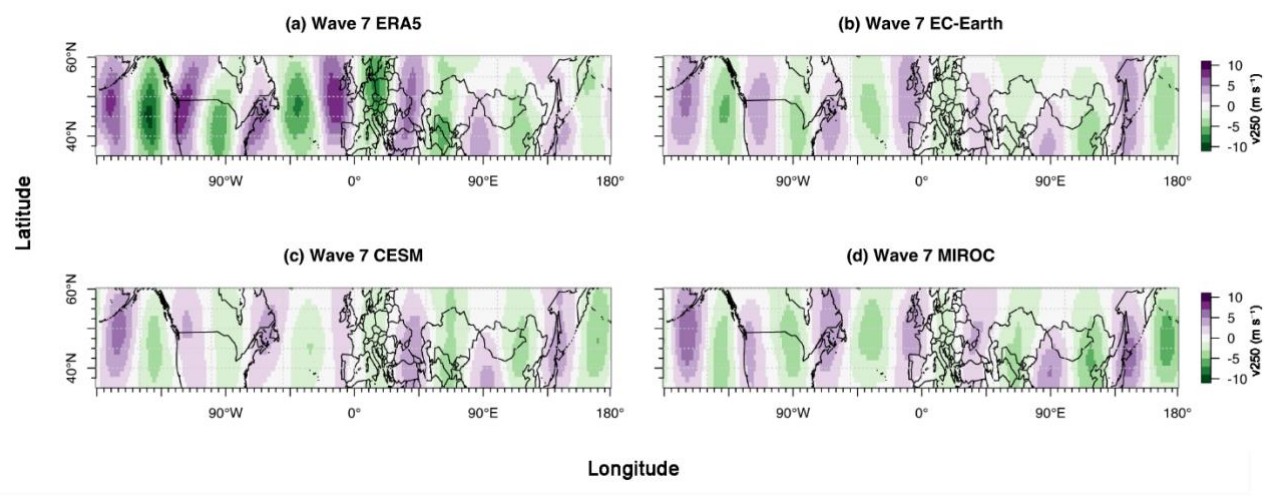

**Figure B11: Composite anomaly plots of weeks with high-amplitude wave-7 episodes for meridional wind velocity at 250hPa (v250, anomaly) in ERA5 (a), EC-Earth (b), CESM (c) and MIROC (d) based on control runs AISI.**

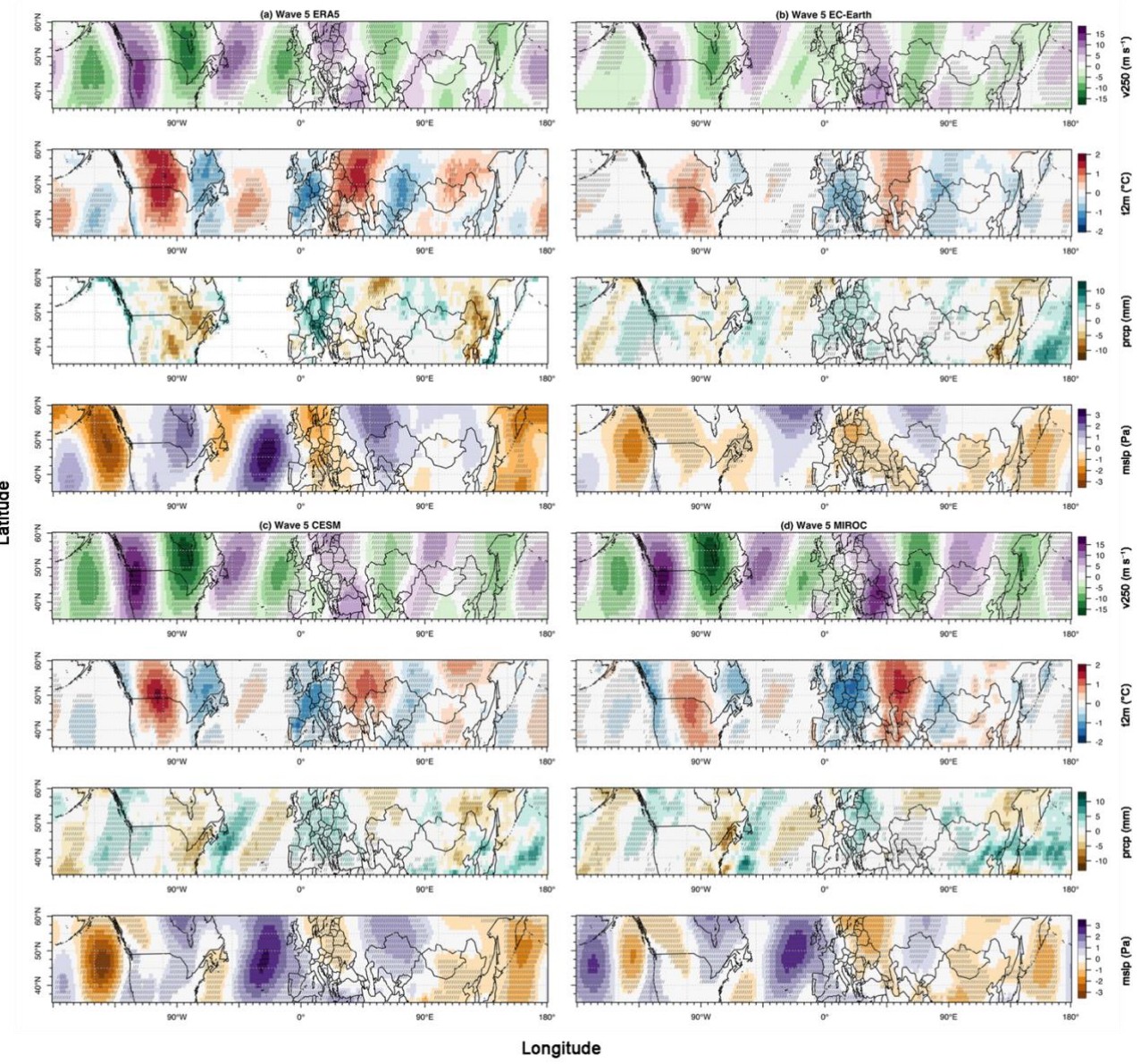

**Figure B12: Same as Fig. 3 but with significance test at 95% confidence level applied. Values significantly exceeding the 95% confidence level from composites between amplified and non-amplified periods are hatched.**

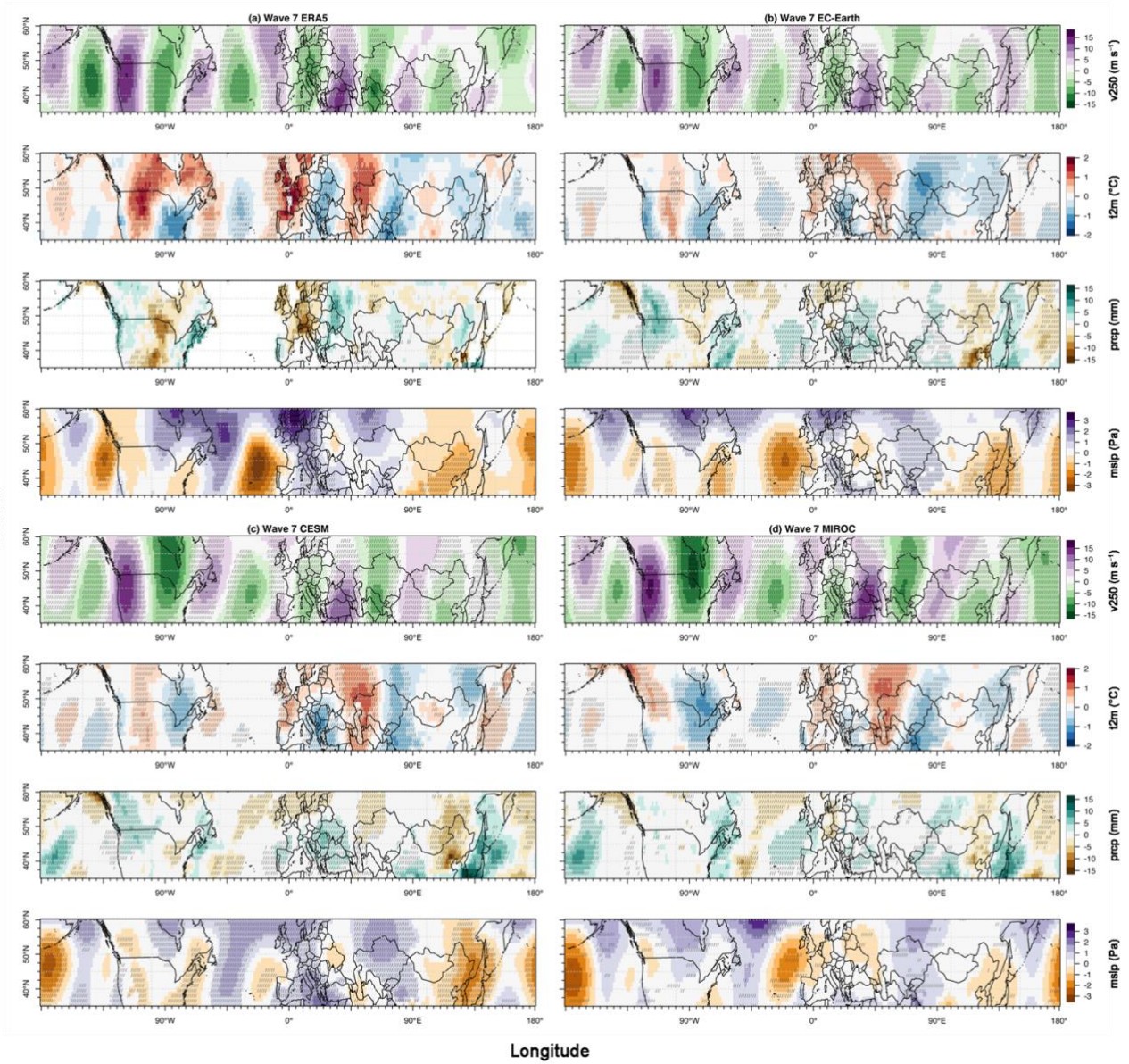


**Figure B13: Same as Fig. 4 but with significance test at 95% confidence level applied. Values significantly exceeding the 95% confidence level from composites between amplified and non-amplified periods are hatched.**
