# Peer review of "Summertime Rossby waves in climate models: Substantial biases in surface imprint associated with small biases in upper-level circulation"

_Weather and Climate Dynamics, 2021_

## Referee Comment (RC1)

Reviewer's comment on the manuscript
"Summertime circumglobal Rossby waves in climate models: Small biases in upper-level circulation create substantial biases in surface imprint" by Luo et al.

General comments

In this manuscript, the authors assess the representation of wavenumber 5 and 7 patterns in three different climate models (EC-Earth, CESM and MIROC), and search for the reason of model biases with respect to the ERA5 reanalysis data. They find that the models represent these wave patterns reasonably well, however, small biases in the upper level circulation lead to large biases in surface variables, like temperature, precipitation and mean sea level pressure. They show a significant improvement of model performance in case the upper level circulation is nudged based on the observed one. In contrast to previous studies, the soil moisture plays only a minor role in the representation of high amplitude wave events, which can be also a consequence of the chosen weekly time scale.

The paper has a very clear structure. The scientific message is clear, useful and very relevant, it can help to improve climate models and produce more realistic future climate scenarios. The language of the paper is understandable, except some (actually, a bit too many for a submitted manuscript…) grammatical and typographical errors. Based on the importance of the message of the paper, I suggest the manuscript to be accepted for publication in WCD, however, only after the authors have taken care of the below mentioned deficits.

My main criticism concerning this work is that it concentrates only on wavenumber 5 and 7 patterns, and there is no critical discussion related to the applied methodology. I understand that previous studies have found these patterns to be relevant for simultaneous extremes, and it is important to test whether the models can represent them or not, but it would increase the scientific value of the paper if the introduction, at least, presents summertime circulation anomalies and related surface extreme events from a broader perspective. At the end, extreme events can be observed during other wavenumbers as well, thus it is important, that models are able to reproduce a wide range of wave numbers not just the mentioned 5 and 7. Although the authors show the spectrum for a range of waves in their results, this is only very shortly discussed in the paper.

In this work, the model experiments are compared to ERA5 reanalysis data, which is supposed to represent reality. Reanalysis data have, however, their biases and deficits too, which should be mentioned in the paper as well. The relatively short period of ERA5 can lead to additional biases of the statistical estimation – it can lead to an under- or over-sampling of certain wavenumbers, as shortly mentioned in the manuscript too. A short discussion of these issues would increase the degree of objectiveness of the paper. A critical evaluation of the used technical tools – Fourier decomposition of the atmospheric field, nudging - in the discussion part would be beneficial as well and would increase the scientific quality.

Overall, this is a good paper with an important scientific message, however, the scientific quality needs to be improved before publication mainly by
1) increasing the accuracy of the wording and presentation of the results,

2) improving the objectiveness related to the used methods and data set, and
3) the work should be put in a broader scientific context.

Specific comments

Figure 2: The probability densities are smoothed. This is not mentioned in the manuscript nor is the smoothing procedure and bandwidth. This information should be included, and the non-smoothed histograms should be shown as well to give a realistic picture of the involved uncertainties.

L47: "*These persistent weather extremes can have disastrous impacts on human health and societies such as wide spread crop failure, infrastructure damage and properties loss, especially when they are defined as compound events.*"
The word "defined" is inappropriate here: The impact is a consequence of the manifestation of a real extreme event, it does not depend on how the event is "defined".

It is not mentioned in the manuscript what kind of observational data set is used for the nudging. This is, however, relevant information, which should be included.

L172: "*ERA5 shows the peak for both the wave amplitude and variance at wavenumber 5*".
According to Fig. 1, the peak in the spectrum of ERA5 is at wavenumber 6. Please clarify this.

It is not clear why in the explanation of the Taylor diagram in L204, the root mean square error is mentioned twice.

L288-L290 "*The observations are complex and location specific as one component within a climate model might be tuned in such a way that it compensates for biases in other components. If so, nudging only one component might not necessarily reduce the overall biases, in this case, prescribing only soil moisture part.*"
The message here is not expressed clearly. It should be rephrased, explained more clearly and in a more elaborated way.

L295-L296: "*…are well captured in different climate models in terms of their climatology, variability, and phase-locking behaviors.*"
The phase locking behavior is captured by the analysed climate models, however I wouldn't call it "well captured" based on what Fig. 2 shows.

L327-L328: "*whereas in our soil moisture prescription experiments the soil moisture was prescribed with values from running the land component driven by atmospheric fields from reanalysis in the model offline, which thus represent much smaller forcings.*"
The formulation is not clear enough, it should be rephrased.

L356: "*increase from 0.71 and 0.99 to 0.63 and 1.06*".
An increase from 0.71 to 0.63? Please clarify.

Technical corrections

L37: "*When applying both soil moisture prescription and the nudging of upper-level atmosphere, both the correlation and n.s.d. values are quite similar to only atmosphere component is nudged experiments.*"
Grammatically incorrect, please rephrase.

L50: "Röthlisberger et al. 2019" instead of "Röthlisbergera et al. 2019".

L57: The citation "Hoskins and Ambrizzi 1993" appears twice.

L72 and L74: "*some have analyzed*" and "*Some studies by Branstator at al…*"
Formulation too general.

L192: "*Then we obtained the occurrences for JJA wave-5 and wave-7 events during 1979 to 2016 for ERA5 are 8.1% and 7.1%*".
Grammatically incorrect.

L212: "Taking ERA5 data as reference" instead of "taking ERA5 data as references".

L283: "completely" instead of "completed"

"And" is not a long word and can be written out, it does not need to be replaced by the symbol "&".

L293 "some extreme events"
Formulation too general.

L340: *"… that persist more than 2 weeks in summer events"*
Please rephrase.

L346: "flow" instead of "slow"

L347: "large-scale circulation pattern" instead of "large circulation pattern"

---

## Author Comment (AC2)

**Manuscript: WCD-2021-48**

Response to the Reviewers' comments on

**Summertime circumglobal Rossby waves in climate models: Small biases in upper-level circulation create substantial biases in surface imprint**

By Fei Luo, Frank Selten, Kathrin Wehrli, Kai Kornhuber, Philippe Le Sager, Wilhelm May, Thomas Reerink, Sonia I. Seneviratne, Hideo Shiogama, Daisuke Tokuda, Hyungjun Kim, and Dim Coumou

We thank both reviewers for their insightful and positive comments. We find all the suggestions constructive, and we are delighted that both reviewers found this article interesting and of scientific importance. We address each comment point by point below. The reviewers' comments are given in **black** and our responses in **blue**.

**Reviewer #1**

General comments

In this manuscript, the authors assess the representation of wavenumber 5 and 7 patterns in three different climate models (EC-Earth, CESM and MIROC), and search for the reason of model biases with respect to the ERA5 reanalysis data. They find that the models represent these wave patterns reasonably well, however, small biases in the upper level circulation lead to large biases in surface variables, like temperature, precipitation and mean sea level pressure. They show a significant improvement of model performance in case the upper level circulation is nudged based on the observed one. In contrast to previous studies, the soil moisture plays only a minor role in the representation of high amplitude wave events, which can be also a consequence of the chosen weekly time scale.

The paper has a very clear structure. The scientific message is clear, useful and very relevant, it can help to improve climate models and produce more realistic future climate scenarios. The language of the paper is understandable, except some (actually, a bit too many for a submitted manuscript...) grammatical and typographical errors. Based on the importance of the message of the paper, I suggest the manuscript to be accepted for publication in WCD, however, only after the authors have taken care of the below mentioned deficits.

My main criticism concerning this work is that it concentrates only on wavenumber 5 and 7 patterns, and there is no critical discussion related to the applied methodology. I understand that previous studies have found these patterns to be relevant for simultaneous extremes, and it is important to test whether the models can represent them or not, but it would increase the scientific value of the paper if the introduction, at least, presents summertime circulation anomalies and related surface extreme events from a broader perspective. At the end, extreme events can be observed during other wavenumbers as well, thus it is important, that models are able to reproduce a wide range of wave numbers not just the mentioned 5 and 7. Although the authors show the spectrum for a range of waves in their results, this is only very shortly discussed in the paper.

We agree with the reviewer that including analyses of more wavenumbers will improve the scientific value of the manuscript. In the revised manuscript we will provide additional figures of waves 4-8 to provide a broader range of wavenumbers relevant for extremes.

In this work, the model experiments are compared to ERA5 reanalysis data, which is supposed to represent reality. Reanalysis data have, however, their biases and deficits too, which should be mentioned in the paper as well. The relatively short period of ERA5 can lead to additional biases of the statistical estimation – it can lead to an under- or over-sampling of certain wavenumbers, as shortly mentioned in the manuscript too. A short discussion of these issues would increase the degree of objectiveness of the paper. A critical evaluation of the used technical tools – Fourier decomposition of the atmospheric field, nudging - in the discussion part would be beneficial as well and would increase the scientific quality.

Overall, this is a good paper with an important scientific message, however, the scientific quality needs to be improved before publication mainly by

1) increasing the accuracy of the wording and presentation of the results,

2) improving the objectiveness related to the used methods and data set, and

3) the work should be put in a broader scientific context.

Thank you for the list and suggestions. Some key aspects of our responses and revisions regard the above suggestions are:

1) We will revise the manuscript carefully to improve the overall accuracy and scientific presentations of the results.

2) We will add one subsection in "Data and Methods" to include the details on the atmospheric nudging including its vertical profile. We will provide a critical discussion on limitations of ERA5.

3.a) For the main concern from Reviewer #1 about presenting that models can reproduce a broader range of different waves numbers and also their related surface extreme events. We decided to add one paragraph on this topic in the introduction part during the revision process.

3.b) We will add one small subsection in "Discussion" to discuss the limitations of the methods we used such as Fourier decomposition of the atmospheric field and nudging to improve this paper's scientific quality. We will also discuss the potential biases in ERA5 data to improve the objectiveness of this paper.

Specific comments

Figure 2: The probability densities are smoothed. This is not mentioned in the manuscript nor is the smoothing procedure and bandwidth. This information should be included, and the non-smoothed histograms should be shown as well to give a realistic picture of the involved uncertainties.

Thank you for this comment, we have included the bandwidth information in the caption in Figure 2 as shown below. We also constructed the histograms for Figure 2.

[Figure]

**Figure 2: Phase-locking of Rossby waves for JJA ERA5 and model waves 5-8 in control run AISI for high wave amplitude events (> 1.5 s.d.):** (a)-(d), Probability density functions of the phase positions of waves 5-8 in ERA5, EC-Earth, CESM, and MIROC during JJA for the period of 1979-2015/2016 (wave 5 (a), wave 6 (b), wave 7 (c), wave 8 (d)). The bandwidth for ERA5 and models are as follows: (a) Wave 5: 0.35(ERA5), 0.40(EC-Earth), 0.25(CESM), 0.22(MIROC), (b) Wave 6: 0.53(ERA5), 0.45(EC-Earth), 0.30(CESM), 0.25(MIROC), (c) Wave 7: 0.25(ERA5), 0.29(EC-Earth), 0.27(CESM), 0.22(MIROC), (d) Wave 8: 0.49(ERA5), 0.39(EC-Earth), 0.52(CESM), 0.46(MIROC).

Below is the figure where the histograms of ERA5 and model data are shown for high amplitude wave-5 to wave-8 phase-locking distributions: (a) wave 5, (b) wave 6, (c) wave 7, and (d) wave 8. We think it's not ideal to plot everything together (i.e. all the model and ERA5 histograms in one figure), as that way one cannot differentiate between datasets anymore, hampering the interpretation of the figure. Thus, we prefer to show the histograms between ERA5 and the models one by one, and we will add this figure in Appendix.

[Figure]

L47: *"These persistent weather extremes can have disastrous impacts on human health and societies such as wide spread crop failure, infrastructure damage and properties loss, especially when they are* *defined* *as compound events."*

The word "defined" is inappropriate here: The impact is a consequence of the manifestation of a real extreme event, it does not depend on how the event is "defined".

Thank you, we agree on the above suggestion and we changed the original phrase to "especially when these extreme events co-occur".

It is not mentioned in the manuscript what kind of observational data set is used for the nudging. This is, however, relevant information, which should be included.

Thank you for this comment. Below is the vertical atmospheric nudging profile taken from the Figure 1 in a recent work by Wehrli et al. (2021, in review). More information on the atmospheric nudging setup and dataset used can be found in this paper. The winds dataset used for atmospheric nudging are from ERA-Interim reanalysis (Dee et al., 2011).

We will add this information in the revised manuscript.

[Figure]

*Figure 1. Nudging profile for the three ExtremeX ESMs. The actual pressure levels are marked with an x and joined with lines. The nudging intensity is given from zero (no nudging) to one (fully nudged) (Wehrli et al., 2021, in review). Taken with permission.*

L172: "*ERA5 shows the peak for both the wave amplitude and variance at wavenumber 5*". According to Fig. 1, the peak in the spectrum of ERA5 is at wavenumber 6. Please clarify this.

Thank you for spotting this, indeed it should be "at wavenumber 6". The typo is corrected accordingly.

It is not clear why in the explanation of the Taylor diagram in L204, the root mean square error is mentioned twice.

Thank you, the second "the centered RMSE" is deleted.

L288-L290 "*The observations are complex and location specific as one component within a climate model might be tuned in such a way that it compensates for biases in other components. If so, nudging only one component might not necessarily reduce the overall biases, in this case, prescribing only soil moisture part.*"

The message here is not expressed clearly. It should be rephrased, explained more clearly and in a more elaborated way.

Thank you, the sentence is rephrased as follows:

"The aforementioned observations are location specific as one component within a climate model might, erroneously, be tuned in such a way that it compensates for biases in other components of the climate model. If so, nudging only that component would not reduce the overall bias. In this case, prescribing only soil moisture part does not guarantee the reduction of overall bias."

L295-L296: "*...are well captured in different climate models in terms of their climatology, variability, and phase-locking behaviors.*"

The phase locking behavior is captured by the analysed climate models, however I wouldn't call it "well captured" based on what Fig. 2 shows.

Thank you for pointing this out, we changed "phase-locking behaviors" to "The phase-locking bevaior is captured."

L327-L328: "*whereas in our soil moisture prescription experiments the soil moisture was prescribed with values from running the land component driven by atmospheric fields from reanalysis in the model offline, which thus represent much smaller forcings.*"

The formulation is not clear enough, it should be rephrased.

Thank you, the sentence is rephrased as follows:

"whereas in our prescribed soil moisture experiments, the soil moisture is not set to zeros. Instead, soil moisture is set to more realistic values coming from the models' land component forced by atmospheric fields from reanalysis. Our experiment thus represents much smaller forcings than prescribing the soil moisture to zero as done in Teng et al. (2019)."

L356: "*increase from 0.71 and 0.99 to 0.63 and 1.06*". An increase from 0.71 to 0.63? Please clarify.

Thank you for spotting this, it should be "increase from 0.71 and 0.63 to 0.99 and 1.06".

Technical corrections

L37: "*When applying both soil moisture prescription and the nudging of upper-level atmosphere, both the correlation and n.s.d. values are quite similar to only atmosphere component is nudged experiments.*"

Grammatically incorrect, please rephrase.

Thank you, the sentence is rephrased as follows:

"When applying both soil moisture prescription and the nudging of upper-level atmosphere, both the correlation and n.s.d. values are quite similar to the experiments where only the atmosphere component is nudged."

L50: "Röthlisberger et al. 2019" instead of "Röthlisbergera et al. 2019".

Thank you, the typo is corrected accordingly.

L57: The citation "Hoskins and Ambrizzi 1993" appears twice.

Thank you for spotting this, the typo is corrected accordingly.

L72 and L74: "*some have analyzed*" and "*Some studies by Branstator at al...*" Formulation too general.

We implied the citations (Garfinkel et al., 2020; Wills et al., 2019) and Branstator et al. (2002 & 2017) as "some have analyzed" and "Some studies by Branstator" here. But we do think the suggestion above is helpful, thus we changed the original sentences as follows:

"Although studies such as Garfinkel et al. (2020) and Wills et al. (2019) have analyzed waves in models, their focus is not on summer and, also, they have not explored the phase-locking behavior of amplified, quasi-stationary Rossby waves. Furthermore, most studies have not analyzed waves above wave number 6. Studies by Branstator et al. (2002 & 2017) have also looked into models but focus on seasonal means and/or winter."

L192: "*Then we obtained the occurrences for JJA wave-5 and wave-7 events during 1979 to 2016 for ERA5 are 8.1% and 7.1%*". Grammatically incorrect.

Thank you, we deleted "Then we obtained" and changed "occurrences for" to "occurrences of". The sentence is now as follows: "The occurrences of JJA wave-5 and wave-7 events during 1979 to 2016 for ERA5 are 8.1% and 7.1%".

L212: "Taking ERA5 data as reference" instead of "taking ERA5 data as references".

Thank you, changed as suggested.

L283: "completely" instead of "completed"

Thank you, changed as suggested.

"And" is not a long word and can be written out, it does not need to be replaced by the symbol "&".

We couldn't find the "&" symbol in L283. Did you mean the symbol "&" in L283 or in general throughout the whole paper?

L293 "some extreme events" Formulation too general.

Thank you, we deleted the word "some", and rewrote as "extreme events such as heatwaves and heavy precipitation".

L340: *"… that persist more than 2 weeks in summer events"* Please rephrase.

Thank you for the suggestion, we rephrased the whole sentence as follows:

"In the summer, when the wave-5 or wave-7 events persist more than 2 weeks, the average reduction in crop production is 4% and even up to 11% on regional level (Kornhuber et al., 2020)."

L346: "flow" instead of "slow"

Thank you for spotting this, the typo is corrected accordingly.

L347: "large-scale circulation pattern" instead of "large circulation pattern"

Thank you, changed as suggested.

**Reviewer #2**

This study compares model errors with a set of nudging experiments using multiple AGCMs. They find nudging the upper-level circulation can significantly reduce surface biases, while nudging soil moisture can't lead to much improvement. This is a very interesting study that definitely deserves to be published. However, the interpretation and presentation may need some extra efforts. Below I list several concerns, mostly minor, for consideration during the revision process.

Thank you for the positive feedback and we will consider the suggestions for our revision process.

Total field or anomalies. I get confused at different parts of the manuscript whether they were talking about the total field or the anomalies of v250. Wavenumber 5 and 7 may be more prominent and longitudinally phase locked in the total field, but it's unclear whether this is also true for the anomalies. I am afraid that for subseasonal variability and extreme events, what matters more is the anomaly, not the total field. I can't help wondering why they chose to only focus on events that project strongly onto wavenumber-5 and -7 of the total field.

Thank you for this comment, we have decided to do the follows in the revision process:

- We will expand to wave Nr. 4 to Nr.8 as we responded to Reviewer #1
- We prefer to work with absolute v250 values rather than anomalies as the former are easier to interpret. Still we will add the same figures using anomalies of v250 to the appendix

It would be far stretched to use the current experiments to address Question 3, whether model biases originate from the atmosphere circulation or land surface- feedbacks. First, prescribing soil moisture does not necessarily mean the model can accurately simulate land-atmosphere feedback. Secondly, it seems that the nudging they applied to the atmosphere circulation (above 700 hPa) is far more than just "the upper atmospheric levels". Therefore, they may need to rephrase the conclusion that "small bias in the upper atmospheric levels can result in big bias in surface weather conditions" (ln 305). See 4. for another concern on drawing this conclusion.

We agree with the reviewer that the current experiments cannot fully conclusively differentiate between the source of biases. Still, our experiments show relatively minor biases in upper level circulation propagate in models to create larger biases in surface variables. For completeness, we add the vertical atmospheric nudging profile which shows that nudging starts around 700hPa but only with a very weak nudging strength. The nudging strength increases gradually in the upward direction and full nudging is only applied above ca. 400hPa. Thus, we will reword 'upper atmosphere' into 'mid-to-upper atmosphere' as it is important to highlight that the planetary boundary layer is free to adjust in the nudged experiments.

Why are the pattern correlation coefficients in Figs. 5,6 all positive? What's the geographical domain used to construct the Taylor diagrams?

Because the Taylor diagram (Taylor 2001) calculates the field correlation coefficients between ERA5 and each model.

It can be seen in Fig.5 and Fig.6 that the spatial patterns are quite similar (highly correlated), thus the correlation coefficients are all positive. It would imply that the model fields are in the opposite phases as ERA5 to have negative coefficients (i.e. v250 has a positive sign in ERA5 whereas in EC-Earth to have a negative sign), which is not the case.

The geographical domain used to construct the Taylor diagram is the same as in Fig.3 and Fig.4.: the full longitudinal band between 35N to 60N.

Why unlike other three models, CESM (Fig.B1) B_land is not much smaller than B_atm? In fact, why are the magnitudes of B_tot, B_atm, B_land and B_res rather similar?

We don't agree on this point, as if we look at Fig. 5 and Fig. 6, B_land is different from B_atm. This is also shown quantitively in the statistical values in Table A1 to A4. Of course, the architect of each climate model is different, thus it's not surprising that models differ on smaller scale.

Significant test for the composite maps in Figs.3,4 is needed.

Thank you for the suggestion, we think this is a good idea and we will implement the significant test for the composite maps in Fig.3 and Fig.4 during the revision process.

The manuscript needs serious editing.

Thank you, we will revise the manuscript carefully during the revision process.

**References**

Dee, D. P., S. M. Uppala, A. J. Simmons, P. Berrisford, P. Poli, S. Kobayashi, U. Andrae, et al. "The ERA-Interim Reanalysis: Configuration and Performance of the Data Assimilation System." *Quarterly Journal of the Royal Meteorological Society* 137 (656): 553–97. https://doi.org/10.1002/qj.828, 2001.

Röthlisbergera, M., Frossard, L., Bosart, L. F., Keyser, D., and Martius, O.: "Recurrent Synoptic-Scale Rossby Wave Patterns and Their Effect on the Persistence of Cold and Hot Spells." *Journal of Climate* 32 (11): 3207–26, doi: 10.1175/JCLI-D-18-0664.1, 2019.

Taylor, K. E.: "Summarizing Multiple Aspects of Model Performance in a Single Diagram." *Journal of Geophysical Research Atmospheres* 106 (D7), doi: 10.1029/2000JD900719, 2001.

Wehrli, K., Luo, F., Hauser, M., Shiogama, H., Tokuda, D., Kim, H., Coumou, D., et al.: "The ExtremeX Global Climate Model Experiment : Investigating Thermodynamic and Dynamic Processes Contributing to Weather and Climate Extremes," no. July: 1–31, doi:10.5194/esd-2021-58, 2021.

---

## Referee Report (RR1)

Reviewer's comment on the revised manuscript
"Summertime circumglobal Rossby waves in climate models: Small biases in upper-level circulation create substantial biases in surface imprint" by Luo et al.

**General comments**

In this manuscript, the authors assess the representation of mainly wavenumber 5 and 7 patterns in three different climate models (EC-Earth, CESM and MIROC), and search for the reason of model biases with respect to the ERA5 reanalysis data. They find that the models represent these wave patterns reasonably well, however, small biases in the upper level circulation lead to large biases in surface variables, like temperature, precipitation and mean sea level pressure.

In my previous comment, I asked the authors to take care of a few deficits of the paper and to improve the scientific quality by
1) increasing the accuracy of the wording and presentation of the results,
2) improving the objectiveness related to the used methods and data set, and
3) the work should be put in a broader scientific context.

I appreciate the effort of the authors in answering and addressing every single comment. I think that, after the first round of revision, the paper improved a lot. The authors extended the scientific context in which they embed their results, and the objectiveness of the work increased by adding a critical discussion of potential drawbacks of the applied methods. **The accuracy of the wording improved as well, however, there is still substantial need for further modifications. Especially, many recently added parts are unclear and contain several grammatical and technical mistakes. In my opinion, although the nature of mistakes is mostly minor, they are almost inacceptable for a revised manuscript. Thus, I ask the authors again to revise the text thoroughly, otherwise I cannot recommend the acceptance of this paper by WCD.**

**Specific comments**

L224-227:

"… *the v250 field is shown in absolute values, together with surface variables in anomalies with respect to climatology mean*"
You mention this a few lines above (L217-218).

"*The same analysis is carried out … during wave 5 and 7 events (see Appendix Fig. B10 and B11)*"
Which analysis exactly? Too unclear, please reformulate.

"*By comparing Fig.3 and Fig.4 to Fig. B10 and Fig. B11, the observation can be obtained that the spatial patterns of v250 do not differ much*"
Instead of writing "comparing Fig. X and Fig. Y" please write what you are comparing exactly in terms of the figure contents. You could write, for example, that you compare the anomalies corresponding to wave 5 and 7 events with the ones obtained for the other wave events. Written in this form, it becomes

easier for the reader to understand your message.
Please rephrase "*the observation can be obtained*".

Overall, please rephrase this recently added part of the paragraph: don't write the same several times, formulate the message in a clear way and be sure that the new text fits well to the rest of the paragraph.

L229-232:

"*Furthermore, Fig. 3 and Fig. 4 with significant tests at confident level of 95%, as well as False Discovery Rate (FDR) method (Benjamini & Hochberg, 1995) were applied in Fig. B12 and Fig. B13.*"
You actually write here that you apply Fig. 3 and 4 in Fig. B12 and B13, which makes no sense. Please reformulate.
"Significance test" instead of "*significant test*" and "confidence level" instead of "*confident level*".

"*Areas with highlighted fuchsia color are the locations passed the significant tests*"
Locations do not pass the significance test, instead the differences over those areas are significant. Additionally, the sentence grammatically incorrect. Please rewrite.

Please rephrase the whole paragraph. You could incorporate it in the rest of the text because there is no new scientific message here, instead it points out that some already presented results are significant.

Fig. B12 and B13: please rewrite captions ("significant applied at…" ??).

Remark regarding emergent constraints: The option of emergent constraints based on observational data should be considered with caution, see for example Sanderson et al., 2021, Earth Syst. Dynam.

**Technical corrections**

L156: Instead of "the weeks are"  "the numbers of weeks are".

L206: "*can be found at Fig. B2*". Please rephrase.

L299: "*Still in For t2m*" ?

L322-323 "the" is missing before "soil moisture part" and "overall bias".

L329: "*wave-7*" appears twice.

L358-359: "*has little effect on the representation of circumglobal waves nor their surface imprint, or, at least not on the anomaly these events produce.*"
"has little effect on the representation of circumglobal waves **and** their surface imprint, or at least **on** the anomaly these events produce."

L367: "*for during*"?

L372: "*Since its on hemispheric scale…*" Incorrect formulation, please rephrase. "Its" is a possessive pronoun.

L373: "*Depending on the stationarity of the dataset, the results can also differ substantially with FFT method.*" What do you mean?

L382-383: Please move "*patterns*" to the end of the sentence.

L392: "flow" instead of "*slow*". I pointed this out in the previous review already.

References: I'm sure it's "Röthlisberger, M." instead of "*Röthlisbergera, M.*". This is also something I mentioned in the previous review as well.

Fig. B2 caption: "Same as" instead of "*Same with*".

Figure captions: "*seal level pressure*" is written instead of "sea level pressure".

---

## Author Response (AR2)

**Manuscript: WCD-2021-48**

Response to the Reviewers' comments on

**Summertime circumglobal Rossby waves in climate models: Small biases in upper-level circulation create substantial biases in surface imprint**

By Fei Luo, Frank Selten, Kathrin Wehrli, Kai Kornhuber, Philippe Le Sager, Wilhelm May, Thomas Reerink, Sonia I. Seneviratne, Hideo Shiogama, Daisuke Tokuda, Hyungjun Kim, and Dim Coumou

We thank both reviewers for their thoughtful and constructive comments. We find most of the suggestions are valid and will help us to improve the manuscript. We address each comment point by point below. The reviewers' comments are given in **black** and our responses in **blue**.

**Reviewer #1**

**General comments**

In this manuscript, the authors assess the representation of mainly wavenumber 5 and 7 patterns in three different climate models (EC-Earth, CESM and MIROC), and search for the reason of model biases with respect to the ERA5 reanalysis data. They find that the models represent these wave patterns reasonably well, however, small biases in the upper level circulation lead to large biases in surface variables, like temperature, precipitation and mean sea level pressure.

In my previous comment, I asked the authors to take care of a few deficits of the paper and to improve the scientific quality by
1) increasing the accuracy of the wording and presentation of the results,
2) improving the objectiveness related to the used methods and data set, and
3) the work should be put in a broader scientific context.

I appreciate the effort of the authors in answering and addressing every single comment. I think that, after the first round of revision, the paper improved a lot. The authors extended the scientific context in which they embed their results, and the objectiveness of the work increased by adding a critical discussion of potential drawbacks of the applied methods. **The accuracy of the wording improved as well, however, there is still substantial need for further modifications. Especially, many recently added parts are unclear and contain several grammatical and technical mistakes. In my opinion, although the nature of mistakes is mostly minor, they are almost inacceptable for a revised manuscript. Thus, I ask the authors again to revise the text thoroughly, otherwise I cannot recommend the acceptance of this paper by WCD.**

We thank the reviewer for the acknowledgment of the improvements we made to the manuscript during the last revision round. The suggestions from the reviewer are much appreciated. We agree with the reviewer that the text should be modified thoroughly to improve the readability of the manuscript and to eliminate the grammatical and technical mistakes in the paper. We invested substantial time and energy in improving these aspects.

**Specific comments**

L224-227:

"… *the v250 field is shown in absolute values, together with surface variables in anomalies with respect to climatology mean*"
You mention this a few lines above (L217-218).

Thank you for spotting this, we have rephrased the rest of the paragraph as follows to improve the readability and clarity of the message.

"*Figures 3 and 4 show the upper-level meridional wind (v at 250 hPa, absolute field), and anomalies of near-surface temperature (t2m), precipitation (prcp), and sea level pressure (mslp) during high-amplitude events in ERA5 (panel a). The same variables are shown for the free-running atmosphere and soil moisture (AISI) experiments using the three climate models (panels b-d). Extended analysis for other wavenumbers revealed that the evaluated models are capable of reproducing the high-amplitude Rossby waves 4 to 8 and their associated surface anomalies reasonably well (Fig. B3 - B5). The results imply that the model is able to reproduce summertime surface anomalies associated to different wavenumber events. Using anomalies for the upper-level meridional winds, in contrast to absolute v250, gives consistent results (compare Fig. 3 and 4 and Fig. B10 and B11, respectively).*"

"*The same analysis is carried out … during wave 5 and 7 events (see Appendix Fig. B10 and B11)*"
Which analysis exactly? Too unclear, please reformulate.

We have rewritten this part to make it clearer as follows:

"*Figures 3 and 4 show the upper-level meridional wind (v at 250 hPa, absolute field), and anomalies of near-surface temperature (t2m), precipitation (prcp), and sea level pressure (mslp) during high-amplitude events in ERA5 (panel a). The same variables are shown for the free-running atmosphere and soil moisture (AISI) experiments using the three climate models (panels b-d). Extended analysis for other wavenumbers revealed that the evaluated models are capable of reproducing the high-amplitude Rossby waves 4 to 8 and their associated surface anomalies reasonably well (Fig. B3 - B5). The results imply that the model is able to reproduce summertime surface anomalies associated to different wavenumber events. Using anomalies for the upper-level meridional winds, in contrast to absolute v250, gives consistent results (compare Fig. 3 and 4 and Fig. B10 and B11, respectively).*"

"*By comparing Fig.3 and Fig.4 to Fig. B10 and Fig. B11, the observation can be obtained that the spatial patterns of v250 do not differ much*"
Instead of writing "comparing Fig. X and Fig. Y" please write what you are comparing exactly in terms of the figure contents. You could write, for example, that you compare the anomalies corresponding to wave 5 and 7 events with the ones obtained for the other wave events. Written in this form, it becomes easier for the reader to understand your message. Please rephrase "*the observation can be obtained*".

We have rewritten this sentence to make it clearer, please see the quoted sentences :

"*Figures 3 and 4 show the upper-level meridional wind (v at 250 hPa, absolute field), and anomalies of near-surface temperature (t2m), precipitation (prcp), and sea level pressure (mslp) during high-*

*amplitude events in ERA5 (panel a). The same variables are shown for the free-running atmosphere and soil moisture (AISI) experiments using the three climate models (panels b-d). Extended analysis for other wavenumbers revealed that the evaluated models are capable of reproducing the high-amplitude Rossby waves 4 to 8 and their associated surface anomalies reasonably well (Fig. B3 - B5). The results imply that the model is able to reproduce summertime surface anomalies associated to different wavenumber events. Using anomalies for the upper-level meridional winds, in contrast to absolute v250, gives  consistent results (compare Fig. 3 and 4 and Fig. B10 and B11, respectively)."*

Overall, please rephrase this recently added part of the paragraph: don't write the same several times, formulate the message in a clear way and be sure that the new text fits well to the rest of the paragraph.

Thank you for the suggestion and we have modified this part to make it clearer, please see the quoted sentences in the first response in the "Specific comments" section.

L229-232:

"*Furthermore, Fig. 3 and Fig. 4 with* significant tests *at* confident level *of 95%, as well as False Discovery Rate (FDR) method (Benjamini & Hochberg, 1995) were applied in Fig. B12 and Fig. B13.*"
You actually write here that you apply Fig. 3 and 4 in Fig. B12 and B13, which makes no sense. Please reformulate.

Thank you, we have deleted the original sentence and wrote the following instead:

*"Furthermore, Fig. B12 and B13 show that composite anomalies for v250, t2m, and mslp are significant for both waves 5 and 7, accounting for False Discovery Rate (FDR; Benjamini & Hochberg, 1995). "*

"Significance test" instead of "*significant test*" and "confidence level" instead of "*confident level*".

Thank you for spotting this, we have changed the wording.

"*Areas with highlighted fuchsia color* are the locations passed the significant tests"
Locations do not pass the significance test, instead the differences over those areas are significant. Additionally, the sentence grammatically incorrect. Please rewrite.

Thank you, we have deleted this part and made it clearer in the caption of the Fig. B12 and B13.

Please rephrase the whole paragraph. You could incorporate it in the rest of the text because there is no new scientific message here, instead it points out that some already presented results are significant.

Thank you for the suggestion, we have rephrased and merged this paragraph with the one directly above it.

Fig. B12 and B13: please rewrite captions ("significant applied at..." ??).

Thank you, we have rewritten the captions as follows:

"

**Figure B12: Same as Fig. 3 but with significance test at 95% confidence level applied. Values significantly exceeding the 95% confidence level from composites between amplified and non-amplified periods are hatched.**

**Figure B13: Same as Fig. 4 but with significance test at 95% confidence level applied. Values significantly exceeding the 95% confidence level from composites between amplified and non-amplified periods are hatched.**

"

Remark regarding emergent constraints: The option of emergent constraints based on observational data should be considered with caution, see for example Sanderson et al., 2021, Earth Syst. Dynam.

Thank you for the comment. We have removed this line referring to emergent constraint.

**Technical corrections**

L156: Instead of "the weeks are" "the numbers of weeks are".

Thank you, changed as suggested.

L206: "*can be found at Fig. B2*". Please rephrase.

Thank you, we rephrased "can be found at Fig. B2" to "are provided in Fig. B2".

L299: "*Still in For t2m*"?

Thank you, this part should be "Still in EC-Earth, for t2m".

L322-323 "the" is missing before "soil moisture part" and "overall bias".

Thank you, changed as suggested.

L329: "*wave-7*" appears twice.

Thank you for spotting this, we deleted the second "wave-7 events".

L358-359: "*has little effect on the representation of circumglobal waves nor their surface imprint, or, at least not on the anomaly these events produce.*"
"has little effect on the representation of circumglobal waves **and** their surface imprint, or at least **on** the anomaly these events produce."

Thank you for the suggestion, changed as suggested.

L367: "*for during*"?

We changed "applied for during" to "included in".

L372: "*Since its on hemispheric scale...*" Incorrect formulation, please rephrase. "Its" is a possessive pronoun.

Thank you, we rephrased "Since its on hemispheric scale…" to "Since the analysis is done on hemispheric scale…".

L373: "*Depending on the stationarity of the dataset, the results can also differ substantially with FFT method.*" What do you mean?

Thank you, we have added new parts of writing to make it clearer (see section 4.2):

*"As with any choice of circulation metrics, our approach based on Fourier analyses of the zonally oriented wave component has its limitations. This approach implies that if a particular wavelength is pronounced in only one part of the hemisphere, this can result in a high-amplitude FFT signal. Thus, high-amplitude waves (as defined by our metrics) do not necessarily have to be circumglobal. They can either result from a circumglobal wave pattern or a pronounced regional wave pattern. Still, the fact that we find pronounced and significant wave-patterns in our composite analyses reveals that those reflect preferred wave positions. In particular, wave 5 and wave 7 are subject to this phase-locking behavior. Whenever those quasi-stationary waves grow in amplitude, they tend to do so in the same longitudinal phase position thereby causing temperature and precipitation anomalies in the same geographical regions. This has been reported before for observational data, highlighting the risks this creates for multiple breadbasket failures (Kornhuber et al, 2020). The prime motivation of our study is to see how well climate models reproduce these waves, and to that end our FFT-based metric is useful."*

L382-383: Please move "*patterns*" to the end of the sentence.

Thank you, changed as suggested.

L392: "flow" instead of "*slow*". I pointed this out in the previous review already.

References: I'm sure it's "Röthlisberger, M." instead of "*Röthlisbergera, M.*". This is also something I mentioned in the previous review as well.

Apologies for the typos, they were corrected during last revision, but due to some unknown version changes in the Word document, these changes are not recorded. Now they are corrected again.

Fig. B2 caption: "Same as" instead of "*Same with*".
Figure captions: "*seal* level pressure" is written instead of "sea level pressure".

Thank you, changed as suggested.

Response letter to Reviewer #2 continues in the next page.

**Reviewer #2**

**General comments**

The current paper investigates the representation of strong "circumglobal wave events" in three different climate models. A particular focus lies on the role of the upper-level flow versus soil moisture for model biases. First it is shown that the three models in a free-running mode do a pretty reasonable job in representing the anomalies associated with these "extreme circumpolar wave" events as compared with reanalyses. Second, the authors use a valuable set of numerical experiments that was performed in connection with the ExtremeX project in order to find out whether this success is primarily related to the correct representation of the upper-level flow or to the correct representation of the soil moisture conditions. As it turns out, the upper-level flow is considerably more important in this analysis.

The topic is highly relevant, the paper is overall well written, and the logic seems straightforward in large parts of the text.

We would like to thank reviewer for acknowledging the importance of this study and the time and effort for reviewing this paper.

However, the longer I think about this paper, the less I understand what I am supposed to learn. Essentially you investigate composites of events, and the events are selected on the basis of the magnitude of one particular zonal wavenumber. Arguably this is a somewhat artificial selection, because it does not necessarily represent any physical/meteorological situation. For instance, if you select a specific wavenumber and the amplitude of this wavenumber is much larger than the amplitude of all other wavenumbers, the situation corresponds to a circumglobal wavetrain. However, if you select a specific large-amplitude wavenumber and at the same time neighboring wavenumbers are of similar (large) amplitude (something which I assume to be rather common), the meteorological situation is more likely to be a large-amplitude Rossby wave in part of the hemisphere with much smaller amplitudes in the rest of the hemisphere. It transpires that the events that you select (based on one specific wavenumber exceeding a threshold) may be a collection of rather different-looking physical situations, and they are not necessarily always associated with a circumglobal wave train. Taken at face value, this would draw into question even the title of your paper, which promises a study of circumglobal Rossby waves. In any case, if you average over many such separate events (through compositing), it is unclear (to me) what this composite is meant to represent.

We thank the reviewer for these considerations. We agree that our FFT-based approach has its limitations, as any choice of circulation metrics. If a particular wavelength is pronounced in only one part of the hemisphere, this can result in a high amplitude FFT signal. So we agree that high amplitude waves (as defined this way) do not necessarily have to be "circumglobal". They can be circumglobal or result from a regional wave pattern with high amplitude. Thus, we acknowledge the reviewer and remove the word "circumglobally" from the title. Still, the fact that we actually find pronounced (and significant) wave-patterns in our composite analyses reveals that those do not reflect 'a collection of rather different-looking physical situations', as suggested by the reviewer. The reason is that in particular wave 5 and wave 7 are subject to phase-locking: Whenever they grow in amplitude they tend to do so in the same longitudinal phase position which means that they cause temperature and precipitation anomalies in the same geographical regions. This has been reported before for observational data, highlighting the risks this creates for multiple breadbasket failures

(Kornhuber et al, 2020). The prime motivation of our study is to see how well climate models reproduce these waves, and to that end our metric is useful. We have updated the manuscript accordingly, explaining our prime motivation and the usefulness and limitations of our FFT-based wave metric (see line 380 - 389).

Possibly I have not fully understood your analysis. But maybe other readers have a similar problem. Therefore, the authors would have to be more explicit and more lucid in their interpretation and work out much more clearly what one really learns from the analysis of this set of simulations.

Thank you for the suggestion.  We refer to our response to the comment above: We have updated the manuscript explaining our prime motivation more clearly as well as the usefulness and limitations of our FFT-based wave metric(see line 380 - 389).

My second issue is somehow related the previous one. A superficial interpretation of your results suggests that soil moisture is really not important; rather, all you need is to get the upper-level flow right. Maybe this is not how the results should be interpreted, but there is the danger that this impression remains (unless your interpretation becomes much more detailed and explicit). In particular, one of the authors (Sonia Seneviratne) has shown multiple times in previous publications that soil moisture is important to determine summer surface temperatures at least over certain areas over the continents. Again, on a superficial level, the current results seem to contradict these earlier results. I would be interested in Sonia Seneviratne commenting on this issue, and the reader would certainly appreciate if you could clarify.

Thanks for highlighting this. In fact, none of our results are questioning the importance of soil moisture in determining summer surface temperatures. What appears to create the confusion is that the *biases* in t2m (as shown in e.g. Fig 7) are much reduced when we nudge the upper-level circulation to the observed state, but are essentially unaffected when nudging the soil-moisture to the observed state. Based on that we conclude that most of the T2m biases originate from biases in upper-level dynamics. Also, our analyses suggest that soil moisture interactions are reasonably well represented in the climate models. We have added these subtleties in the interpretation of our results with respect to the role of soil moisture in our manuscript, explicitly stating that our results do not question the importance of soil moisture in driving near-surface summer temperatures.

Please see in line 364-377 in the manuscript:

*"To be clear, our results are not questioning the importance of soil moisture as a prime driver of summer surface temperature 365 extremes in various regions throughout the mid-latitudes. Rather, our study shows that prescribing the soil moisture in the models has little effect on surface variables and upper-level variables during high-amplitude wave events. Several studies have shown that soil moisture can play an important role in maintaining large-scale circulation anomalies associated with extremely warm and dry conditions (e.g. Erdenebat and Sato, 2018). In particular, under future climate change reduced soil moisture can lead to a higher probability of heatwaves in Europe during summer via interactions between the land surface and the 370 atmosphere (e.g. Seneviratne et al., 2006). This, however, does not say anything about the imprint of soil moisture biases on biases in near-surface climate in relation to the Rossby wave events investigated in our study. Apparently, it is the state of the atmosphere, i.e. circulation or clouds and precipitation, that governs the model biases in near-surface climate and not so the state of the land surface. Further, in the interpretation of our results, one should be wary that in our prescribed soil moisture runs (AISF/AFSF), we prescribe the 'approximately observed' soil-moisture conditions (and not e.g. soil moisture 375 climatology). This implies that the turbulent heat fluxes in AISF/AFSF still depend on this prescribed soil moisture*

*condition. This means that during for example the heatwave period in Russia in 2010, the prescribed soil moisture will be anomalously dry, which will result in strong sensible heat fluxes. "*

In my interpretation, your analysis points to a possible model bias in the sense that the models systematically under- or overestimate the spectral power in specific wavenumbers as far as the upper tropospheric circulation is concerned. In your analysis this is somewhat obscured by the fact that you focus on just a few wavenumbers. In reality, the lack of spectral power at a certain wavenumber may reappear as a surplus of spectral power at some other wavenumber. In other words, your selection of events is based on highly incomplete information in spectral space and may, therefore, obscure what is really going on. A more straightforward approach to analyse spectral model biases would be to consider the composite spectral power of all wavenumbers and determine related biases. In addition, to the extent that the meteorology in the lower troposphere is "slaved" to the dynamics in the upper troposphere, nudging the upper troposphere makes obviously a difference, but nudging the lower troposphere does not; this would imply that your results regarding the biases are somewhat trivial.

What's non-trivial and what I find highly interesting is the fact that individual wavenumbers apparently have a preferred phase. However, this is not the topic of the current paper, and the current paper does not (aim to) further contribute towards an explanation.

We thank the reviewer for this comment. We were surprised to read that the reviewer concludes that the 'preferred phase' positioning would not be the topic of our paper: Preferred phases, or phase-locking, of individual wavenumbers is actually the prime motivation of our analyses. We attribute this miscommunication to unclear formulations from our side in the earlier version of this manuscript.

We agree with the reviewer that we don't provide new explanations for the underlying physical mechanisms, but this is the first time that this phase-locking behavior is documented in several climate models. The scope of this paper is to identify if climate models are able to capture the characteristics (in terms of mean and variability) of a wide range of wavenumbers, including preferred phase positions. We focus on wave-5 and wave-7 as those were identified to be important in previous literature based on reanalyses (i.e.Kornhuber et al., 2019 & 2020; Drouard et al., 2019)..

We have updated the manuscript explaining our prime motivation in detail and the usefulness and limitations of our FFT-based wave metric (see also our response to the first comment).

Last, but not least, I have an issue with the title, and this mirrors the points that I raised above. I think the title is misleading for two reasons. (1) It suggests that you deal with "circumglobal Rossby waves", which I would argue is not true. You select events on the basis of highly incomplete information in spectral space, and this does not guarantee that your events are characterized by a circumglobal Rossby wave (see my earlier remarks). (2) You say that "small biases in upper-level circulation create substantial biases in surface imprint". To me as a reader this suggests that if you select a specific event and introduce just a small upper-tropospheric bias in that event, this results in strong changes in the surface meteorology. But again, this is not what you have shown. The concept "bias" in your analysis represents a comparison between two composites, e.g., one from a free-running model and the other one from reanalysis data. However, the underlying individual events (from which the composites are computed) may be very different and the analysis, hence, does not support the claim raised in the title (the way I understand it). I realize that you tried to summarize the whole content of your paper in the title, but with an analysis as complex as yours this is invariably going to be impossible.

We also removed the word "circumglobal" and changed the title into: "Summertime Rossby waves in climate models: Substantial biases in surface imprint associated with small biases in upper-level circulation." We hope this satisfies the reviewer.

In summary, I am aware that the current paper uses a quite special (and maybe somewhat artificial and unphysical) setup for the analysis: namely compositing only events that have a large amplitude of a specific wave number. This very special sort of analysis may have a strong imprint on its interpretation, but it is not straightforward (for me) to see and understand this imprint. As a consequence, I am not fully able to follow your interpretations and to appreciate the merits of your analysis.

Thanks for all comments which have certainly improved the quality of our manuscript. We do not agree with the terms "artificial and unphysical", but rather our approach has its pros and cons. We feel confident that in our updated manuscript the pros (i.e. the usefulness of this metric when studying phase-locked behaviour) and cons (i.e. the limitations of the metric) are better described with a much clearer interpretation of our findings (i.e. to do with the role of upper-atmospheric biases and the role of soil moisture). Our study is in line with previous work (Kornhuber et al., 2020) showing the importance of phase-locked waves, and that, to first order, climate models capture such wave events. This is an essential stepping stone for further research into the underlying physics of such wave events.

**Minor issues**

Line 50: arguably, RRWPs are not the same thing as quasi-stationary Rossby waves, so the former should not be given as "an example" for the latter.

We have rephrased the sentence "Persistent weather extremes are often induced by quasi-stationary Rossby waves." to "Persistent weather extremes are often induced by certain Rossby wave patterns."

Line 88: "with the duration of sfc weather conditions….": that's a somewhat strange formulation. Do you mean "long duration" or "persistence" here?
Thank you, we changed the word "duration" to "persistence".

Line 137: "This added tendency term….": this sentence is not clear to me, as well as the ensuing sentence.

We have rephrased the sentence as follows:

"By taking the differences between the model simulations and reference dataset, the added tendency term is computed."

Line 152: Since your analysis is based on weekly averages, you could (and should) say that you are interested in rather persistent anomalies. For instance, in the abstract you could talk about

"summertime persistent (or: quasi-stationary) wave events" instead of just "summertime wave events".

Thank you for the suggestion. We have added some sentences in the abstract accordingly.

Line 157: What is an "imprint"? Please define! Here you talk only about surface "imprints", but in the following plots you also show composite upper tropospheric meridional wind. How are the latter defined?

Imprint in our case means the characteristics, either absolute or abnormal values, of certain variables under certain conditions. In our study, this condition is high-amplified event for wave 5 and wave 7 where the wave amplitudes exceed 1.5 standard deviations from wave climatology. Thus, this term is not strictly tied only to surface variables.

Line 168: The concept of a "bias" is central to your paper (the word appears even in the title), but the "definitions" that you provide on page 6 are not clear to me. In my understanding, AISI represents a free running model run, and ERA5 represents a dataset. So what is the difference between a model run and a data set? This is way too sloppy, this must be specified much more clearly. (You probably mean the difference between the composites, not the difference between model runs or data sets, but this must be said explicitly).
Thank you for the suggestion. We have specified more in the bias definition part.

Line 190: better "variance in wave amplitude" instead of "variance in wave activity", because "wave activity" has a special meaning in dynamical meteorology (which you do NOT want to refer to here).

Thank you for the suggestion, we agree with the reviewer and changed accordingly.

Caption Fig 2: not clear to me what the term "bandwidth" refers to here.

This information was requested before by another reviewer (See Fig. B2). The bandwidth is associated with a histogram and varies with different numbers of bins.

Line 204: Well, ERA5 shows a strong preference for one part of the hemisphere as opposed to the other part of the hemisphere. Therefore, since you have not clearly defined "phase locking", it is not obvious to me that ERA5 is supposed to not show phase-locking for wave-6 and 8.

We do not agree with the reviewer's comment, as we have defined phase-locking in the paragraph above in manuscript as "*a single symmetrical peak in the probability density function*".

Lines 229/230: english? In addition, I appreciate your effort to determine statistical significance, but I was not able to extract from Fig. B12 and B13 what areas are significant. Please clarify. Where on the plots should I see "highlighted fuchsia color"?

We have changed the whole paragraph and also the Fig. B12 and B13 as follows to eliminate the potential misunderstandings:

*"Figures 3 and 4 show the upper-level meridional wind (v at 250 hPa, absolute field), and anomalies of near-surface temperature (t2m), precipitation (prcp), and sea level pressure (mslp) during high-*

*amplitude events in ERA5 (panel a). The same variables are shown for the free-running atmosphere and soil moisture (AISI) experiments using the three climate models (panels b-d). Extended analysis for other wavenumbers revealed that the evaluated models are capable of reproducing the high-amplitude Rossby waves 4 to 8 and their associated surface anomalies reasonably well (Fig. B3 - B5). The results imply that the model is able to reproduce summertime surface anomalies associated to different wavenumber events. Using anomalies for the upper-level meridional winds, in contrast to absolute v250, gives consistent results (compare Fig. 3 and 4 and Fig. B10 and B11, respectively)."*

Line 293: "almost fully…." seems somewhat overstated, I would say "to a large extent…."

Thank you for the suggestion, we changed the phrase as suggested above.

Line 304: Ref to Fig 7a seems wrong.

Thank you, it should be Fig. 5(a).

Line 316: Ref to Fig 5b seems wrong.

This is correct reference.

Line 329: what do the correlation values given in parentheses refer to?

There is only one reference used in this paper, namely ERA5, thus throughout the paper it's always the model data compared to ERA5.

Line 330: better "climatology and variability".

Thank you, changed as suggested.

Line 343: I think that the vertical wind in your nudging experiments is only weakly constrained by the divergence/convergence of the horizontal wind. Rather, having the horizontal wind almost right in these experiments implies the correct forcing in the omega equation and, hence, a good representation of the vertical wind. In other words, I think your statement is correct, but the reason/explanation you give may not be correct.

We did propose the explanations as potential mechanisms.

Line 347: "can be disturbed in the models….": that's unclear. I agree that an error in the vertical wind may lead to an error in cloudiness, but there may be other sources for errors in cloudiness (e.g., moisture advection) which may be just as important.

We agree with the reviewer.

Line 363: froced -> forced

Thank you, the typo is corrected accordingly.

Line 371/372: do you mean that they yield cirumglobal waves "by design"!?

No, that is not what we meant. We have explained the FFT approach in the response letter in "General comments".

Line 373: What is the "stationarity of a data set"? also: I hope very much that the results do NOT depend on the method used to compute the FFT! Can you explain what you mean here?

No, the results do not depend on the method used to compute the FFT. We intended to point out some limitations of the methods and to have some critical discussions.

Line 374: I do not understand this. If you provide only the zonal mean to an FFT algorithm, then the wave amplitude would be zero for all zonal wavenumbers except s=0.

We have rewritten the whole part to make it clearer, see in line 380 – 389.

Line 376: not "the data is bigger….", rather "the amount of data is larger by a factor…."

Thank you, changed as suggested.

Line 392: "slow to…."??? Also: a simple statistical connection between the upper-level flow and surface temperatures per so is not an "emergent constraint" (the way I understand that concept). Please clarify.

It should be "flow" instead of "slow". We have removed the part "emergent constraint" to avoid misunderstandings.

We also would like to thank the editor sincerely for the constructive and insightful comments, which we have considered when revising this paper.

**References:**

Drouard, M., Kornhuber, K., & Woollings, T.: "Disentangling dynamic contributions to summer 2018 anomalous weather over Europe." *Geophysical Research Letters* (46): 12,537–12,546. doi: 10.1029/2019GL084601, 2019

Erdenebat, E., & Sato, T.: "Role of soil moisture-atmosphere feedback during high temperature events in 2002 over Northeast Eurasia. " *Progress in Earth and Planetary Science*, *5*(1), 1-15, doi: 10.1186/s40645-018-0195-4, 2018.

Kornhuber, K., Osprey, S., Coumou, D., Petri, S., Petoukhov, V., Rahmstorf, S., and Gray, L.: "Extreme Weather Events in Early Summer 2018 Connected by a Recurrent Hemispheric Wave-7 Pattern." *Environmental Research Letters* 14 (5): 054002, doi: 10.1088/1748-9326/ab13bf, 2019.

Kornhuber, K., Coumou, D., Vogel, E., Lesk, C., Donges, J. F., Lehmann, J. and M. Horton, R.: "Amplified Rossby Waves Enhance Risk of Concurrent Heatwaves in Major Breadbasket Regions." *Nature Climate Change* 10 (1): 48–53, doi: 10.1038/s41558-019-0637-z, 2020.

Seneviratne, S., Lüthi, D., Litschi, M., and Schär, C.: "Land–atmosphere coupling and climate change in Europe." *Nature*: 443, 205–209, doi: 10.1038/nature05095, 2006

---

## Author Response (AR3)

Manuscript: WCD-2021-48

Response to the Reviewer' comments on

**Summertime Rossby waves in climate models: Substantial biases in surface imprint associated with small biases in upper-level circulation**

By Fei Luo, Frank Selten, Kathrin Wehrli, Kai Kornhuber, Philippe Le Sager, Wilhelm May, Thomas Reerink, Sonia I. Seneviratne, Hideo Shiogama, Daisuke Tokuda, Hyungjun Kim, and Dim Coumou

We thank the reviewer for reviewing the paper again and is satisfied with our efforts and revisions from last round. We find most of the suggestions are constructive and will help us to finalize the manuscript. We address each comment point by point below. The reviewer's comments are given in **black** and our responses in **blue**.

**Reviewer #1**

**General comments**

This is the revised version of a manuscript that I have reviewed before. My original concerns have been addressed to a large extent by the reply to my earlier comments as well as by the revisions.

The paper sort-of "validates" three state-of-the-art climate models in terms of the statistical behavior of selected episode during which a few selected Fourier components of the upper tropospheric meridional wind have strong amplitude. The motivation for those specific Fourier components stems from an earlier paper in which the authors have shown an interesting connection to summer heat extremes. The main point here is that the models represent these large-amplitude wave-5 and wave-7 events sort of OK in terms of the upper tropospheric circulation statistics, while there is a substantial underrepresentation (and partly misrepresentation) in terms of associated surface variables. The situation is analyzed in more detail with the help of model simulations in which certain model variables are nudged towards observed conditions. In particular, the underestimation close to the surface almost disappears when the upper tropospheric flow is nudged towards observations.

The analysis seems to be sound, the topic is relevant, and the results are intriguing. I have a few further issues which should be taken into account when preparing the final version of this manuscript.

We thank again the reviewer for the acknowledgment of the improvements we made to the manuscript during the last round of revision. We are delighted to know that the reviewer found our results intriguing and also gave a nice summary of the essence of this paper.

**Major issues**

1.) The selection of episodes (both in reanalyses and in the free-running model simulations) is based on the large amplitude of a specific wavenumber of the upper tropospheric meridional wind. This procedure does not imply that the episodes in the reanalyses and in the model simulations represent the same "events". The use of the term "event" is, therefore, somewhat misleading.

The situation changes fundamentally when the upper tropospheric flow in the model simulations is nudged to the reanalyses: now the different episodes, indeed, represent the same (or at least very similar) events. In particular, the associated surface anomalies now turn similar to the observed ones (from reanalyses) to the extent that the upper tropospheric flow determines the lower tropospheric meteorology, which is true to a considerable extent.

I think it would be helpful to make a clear distinction between "episodes" and "events".

We thank the reviewer for this comment, and we agree with the reviewer that the distinction between "episodes" and "events" should be made in the manuscript. We have changed "events" to "episodes" when referring to free-running model experiments and followed the terminology throughout the whole manuscript. Thus, for AISI and AISF runs, we use the term "episode" whereas for AFSI and AFSF runs, we use the term "event".

2.) Although the biases of the free-running EC model in terms of the upper-tropospheric flow are, in my eyes, quite substantial, they are referred to by the authors as "minor" or "small". I think it would be fair to acknowledge that they are not just "minor" or "small". This, in combination with the fact that the composite from the free-running model simulations is not composed of the same "events" as the composite from the reanalyses, renders the result somewhat less surprising: namely that nudging the upper tropospheric circulation improves the situation substantially.

We thank the reviewer for mentioning this point. We agree that the usage of "minor" or "small" can be somewhat subjective, thus we tried to eliminate subjective wording during this round of revision when referring to the biases in upper-level circulation (v250) and put the terms in context when needed. For example, our findings show that the biases in upper-level atmospheric flow is smaller compared to those at the surface.

**Minor issues**

Line 156, 209, 370: What do you exactly mean by the term "imprint"? Does it suggest a causal relationship? It seems that the latter is unproven at this point.

No, it does not imply a causal relationship and we also do not imply causality in the manuscript. We have also explained the term "imprint" in our response letter from last revision. See below:

*"Imprint in our case means the characteristics, either absolute or abnormal values, of certain variables under certain conditions. In our study, this condition is high-amplified event for wave 5 and wave 7 where the wave amplitudes exceed 1.5 standard deviations from wave climatology. Thus, this term is not strictly tied only to surface variables."*

Line 605, caption of Fig 1: if the whiskers indicate the maximum and minimum values that have occurred, then all values are taken care of, right? In this case there should be no "outliers that are not shown in the plot". Have I missed something?

Thank you for spotting this, indeed there is some confusion here that needs to be explained better. Between upper and lower bars is the 1.5 times interquartile range (1.5 IQR).

The caption of Fig.1 has been modified as follows:

**Figure 1: Boxplot for wave amplitudes in AISI climatology runs for climate models EC-Earth, CESM, and MIROC, as well as reanalysis data ERA5 for the period of June, July, and August in 1979-2015/2016. Red dots indicate the mean, and thick black lines represent the median. The lower hinge of each box is Q1 quartile (25th), and the upper hinge for Q3 quartile (75th). The upper and lower bars are based on the 1.5 times interquartile range (IQR) value. The outliers are not shown in the plot.**

Below is the same as Fig.1 but with outliers plotted.

[Figure]

Line 210: this sentence is somewhat non-sensical: even if the wave gets phase-locked in a not-preferred position, I would expect a prolongation of surface warm anomalies.

Thank you for this comment. We agree with the reviewer and have removed this whole sentence.

Line 212: Here you talk about "events", suggesting that your method singles out specific meteorological situations. However, as argued above as well as in my previous review, this is not the case, rather your selection of "events" may contain a collection of rather different-looking meteorological situations. Therefore, the use of the term "event" is misleading. Rather, you single out specific "episodes", not specific "events".

Thank you for this comment. We agree with the reviewer and changed "events" to "episodes". Please also see our response in the section "Major issues" part 1).

Line 264 ff: The episodes that you study are selected using an upper-tropospheric criterion. For me, it therefore does not come as a surprise that the associated "bias" in the upper troposphere is less than in the lower troposphere. Can you comment, i.e., give the reader some guidance at this point?! Moreover, as stated above, I do not really find the biases in the upper troposphere to be "minor" or "small".

We basically agree that this is not necessarily surprising, it's simply the findings of our work. For the wording issue of "minor" or "small", please see our response in the section "Major issues" part 2). We have also rewritten the sentence, see below:

*"One common finding from both wave-5 and wave-7 events and episodes is that the biases (n.s.d. ≥ 0.75) in upper-level circulation are smaller compared to more-pronounced biases in the surface anomalies t2m, prcp, and mslp."*

Line 269: these two figures do not show biases, but composite anomalies. The biases can, at best, be inferred from these plots.

We agree with the reviewer and have rephrased the original sentence as follows:

*"Next, we aim to infer the biases from composites of anomalies shown in Figs 3 and 4 for the upper-level wind and surface fields."*

Line 285: unclear what the word "this" refers to.

We agree with the reviewer. We have rewritten the sentence as follows:

*"As for wave 5, the free-running EC-Earth (AISI) ..."*

Line 326: "individual Fouriermodes" would be better that "amplified Rossby waves".

Thank you, changed as suggested.

Line 333: again, you should not talk about "events" here (as detailed above). I you were more precise and talk only about "episodes", the result (namely that surface variables do not necessarily show concomitant anomalies) would appear somewhat less surprising.

Similarly, the fact that these "biases" disappear when the (fairly substantial!) upper tropospheric biases are removed through nudging, turns less surprising, because you make a transition to the same "events" (as defined through upper tropospheric dynamics").

Thank you for the suggestion and comments, please see our response above in the section "Major issues" part 1) and 2).

Line 344: "... too weak in the models": why?

Thank you for the question. We have added one sentence after "... too weak in the models." See below:

*"As this subsidence might not be well represented in the models, because of biases in the upper-level flow."*